# Murine interfollicular epidermal differentiation is gradualistic with GRHL3 controlling progression from stem to transition cell states

Ziguang Lin[1], Suoqin Jin [2,3,4], Jefferson Chen[1], Zhuorui Li[1], Zhongqi Lin[1], Li Tang[1], Qing Nie[2,3,4 ✉] & Bogi Andersen [1,5 ✉]

The interfollicular epidermis (IFE) forms a water-tight barrier that is often disrupted in inflammatory skin diseases. During homeostasis, the IFE is replenished by stem cells in the basal layer that differentiate as they migrate toward the skin surface. Conventionally, IFE differentiation is thought to be stepwise as reflected in sharp boundaries between its basal, spinous, granular and cornified layers. The transcription factor GRHL3 regulates IFE differentiation by transcriptionally activating terminal differentiation genes. Here we use single cell RNA-seq to show that murine IFE differentiation is best described as a single step gradualistic process with a large number of transition cells between the basal and spinous layer. RNA-velocity analysis identifies a commitment point that separates the plastic basal and transition cell state from unidirectionally differentiating cells. We also show that in addition to promoting IFE terminal differentiation, GRHL3 is essential for suppressing epidermal stem cell expansion and the emergence of an abnormal stem cell state by suppressing Wnt signaling in stem cells.

[1] Department of Biological Chemistry, School of Medicine, University of California, Irvine, CA, USA. [2] Department of Mathematics, University of California, Irvine, CA, USA. [3] Department of Developmental & Cell Biology, School of Biological Sciences, University of California, Irvine, CA, USA. [4] NSF-Simons Center for Multiscale Cell Fate Research, University of California, Irvine, CA, USA. [5] Department of Medicine, School of Medicine, University of California, Irvine, CA, USA. ✉email: qnie@math.uci.edu; bogi@uci.edu

During development, the epidermis emerges from the surface ectoderm as a single layer of unspecified progenitor cells, which later form the stratified interfollicular epidermis (IFE), the hair follicles, the sebaceous glands, and the tactile sensing Merkel cells. Meanwhile, mesoderm-derived immune cells and neural crest-derived melanocytes take up residence in the epidermis. Although IFE heterogeneity and differentiation have been previously studied with single cell RNA-sequencing (scRNA-seq) in adult mouse and human skin[1–5], late development of the embryonic IFE has not been studied with scRNA-seq, and key gene regulators of IFE differentiation have not been systematically identified with this method.

Conventionally, IFE differentiation is thought to be stepwise as reflected by the IFE's four distinct layers separated by clear boundaries. Cells of the basal layer, which rest on the basal lamina, are mitotically active stem cells marked by the expression of keratin (K) 5 and K14. As the cells move to the spinous layer, they exit the cell cycle; K5 and K14 expression is replaced by K1 and K10 expression. And as the cells advance to the granular layer, they turn on the expression of barrier-forming genes, including loricrin and filaggrin[6]. Eventually, the terminally differentiated cells cease transcriptional activities and die, flattening and de-nucleating to form the watertight cornified layer that is eventually shed from the skin surface. The layered structure of the epidermis suggests that molecular switches could sharply alter the expression of many genes at the boundaries of distinct cell layers. This prediction, however, has not been studied at a genome-wide scale in the postnatal day (P) 0 mouse epidermis, which—similar to the human epidermis, but in contrast to the adult mouse epidermis—contains clearly demarcated epidermal layers.

Layer-restricted expression of transcription factors is thought to contribute to the IFE differentiation program. One such transcription factor, GRHL3, is expressed throughout the developing epidermis during embryogenesis, but becomes restricted to the differentiated layers at P0. Grhl3-deleted mice have a defective epidermal barrier with decreased expression of cell–cell adhesion molecules, lipid producing enzymes, and proteins required for cornified envelope formation and crosslinking[7,8]. At P0, the Grhl3$^{-/-}$ IFE contains a disorganized basal layer, thickened spinous and granular layers, and a compacted cornified layer. Based on these findings, it has been assumed that the main embryonic role of GRHL3 is to activate genes required for full differentiation of cells of the granular layer. But the nature of the IFE hyperplasia in the P0 Grhl3$^{-/-}$ mice remains enigmatic.

To better understand epidermal differentiation, we investigate single cell transcriptomes from mouse skin during embryogenesis and up to P0 in wild type (WT) and Grhl3$^{-/-}$ mice. Our findings challenge the classical notion of a stepwise IFE differentiation, which assumes that cells within a layer are relatively uniform but undergo dramatic changes as they move to the layer above. Rather, we find a high proportion of transition cells with a character intermediate between the basal and the first spinous layer, as well as other features suggesting that IFE differentiation is best viewed as a single-step gradualistic process. RNA velocity analysis, though, indicates that prior to the transition-differentiation cell state boundary, cell states are plastic, whereas after this commitment point, cells states proceed strongly in a unidirectional manner toward terminal differentiation. As expected, we find defective activation of terminal differentiation in Grhl3$^{-/-}$ mice. But unexpectedly, we find accumulation of epidermal stem cell populations and the emergence of proliferative cell states unique to the mutated epidermis. We show that the aberrantly expanded stem cell compartment exhibits increased Wnt signaling while the suprabasal cells exhibit reduced Wnt antagonist expression, with GRHL3 directly binding to key Wnt signaling components. Thus, GRHL3 plays an important role in tempering Wnt signaling and expansion of IFE stem cells during epidermal differentiation.

## Results

### scRNA-seq reveals newborn mouse epidermal cell heterogeneity.
We started our study into IFE differentiation at a single cell level by focusing on the WT P0 IFE. At this stage, the IFE has reached its maximum thickness with morphologically well-defined layers: basal, spinous, granular, and cornified. We generated single cell transcriptomes from the back epidermis, capturing 5494 cells with 38,879 mean number of reads per cell and 2388 mean number of genes per cell. Clustering identified 16 subpopulations of epidermal cells (Fig. 1a). Each cluster was annotated by marker genes that are known to be uniquely expressed in each cell type or cell state (Fig. 1b; Supplementary Fig. 1A–C). We identified all previously defined epidermal subpopulations of the adult epidermis[2]: IFE, hair follicles, sebaceous gland cells, Langerhans cells, T cells, melanocytes, and Merkel cells (Fig. 1a, b). Four adjacent clusters of 1779 IFE cells were identified: two basal clusters (IFE.B1 and IFE.B2; 1002 cells), a basal-suprabasal transition cluster (IFE.T; 350 cells), and a differentiated cluster (IFE.D; 427 cells). The population of transition cells (IFE.T) is 20% of the all IFE cells, which is a surprisingly large fraction. The Gene Ontology (GO) category enrichment[9,10] of the marker genes of the IFE subpopulations indeed reflects the biological functions of each population (Supplementary Fig. 1D–F).

### The IFE consists of numerous basal-spinous transition cells.
To gain greater insights into the progression of IFE differentiation at P0, we next applied pseudo-temporal analysis. We used Monocle[11], an algorithm that uses changes in gene expression between cells to learn in an unbiased manner the position of each cell in a biological trajectory. Monocle introduces the concept of pseudotime as an abstract measure of how far a cell has progressed in a process—here in the differentiation of an IFE stem cell in the basal layer to a terminally differentiated keratinocyte at the top of the epidermis. Monocle ordered the IFE cells along a linear trajectory without introducing significant bifurcations (Fig. 1c). This differentiation trajectory corresponds well with our cluster annotation from Fig. 1a in that the cells progress from IFE.B through IFE.T to IFE.D. To validate the pseudotime ordering, we assessed the expression of canonical markers in the trajectory (Fig. 1d), finding that K14/K5/Col17a1/Itga6 mark cells in the early trajectory, K10/K1/Klf4/Tgm3 in the middle-to-end trajectory, and Lor in the end of the trajectory. These results suggest that Monocle ordered the IFE keratinocytes in a biologically meaningful manner from the most undifferentiated to the most differentiated cells.

We noted high number of transition cells with falling K5/K14 expression and rising K1/K10 expression—characteristics that are intermediary between the basal and spinous layers. This finding is unexpected because of the apparent sharp boundary between the basal and spinous layer[12]. To validate the existence of the transition cells, we performed RNA FISH with Krt14 and Krt10 probes. Consistent with the scRNA-seq data, we observed a substantial number of double positive transition cells both in the basal and in the first spinous layer (Fig. 1e). These results indicate that (1) epidermal differentiation has already begun in the basal layer[13], (2) some spinous cells still retain basal characteristics, and (3) a relatively large population of transition cells exists at the basal-spinous boundary.

### The IFE differentiates gradually in a single-step.
To better understand differentiation at a molecular level, we identified 4299

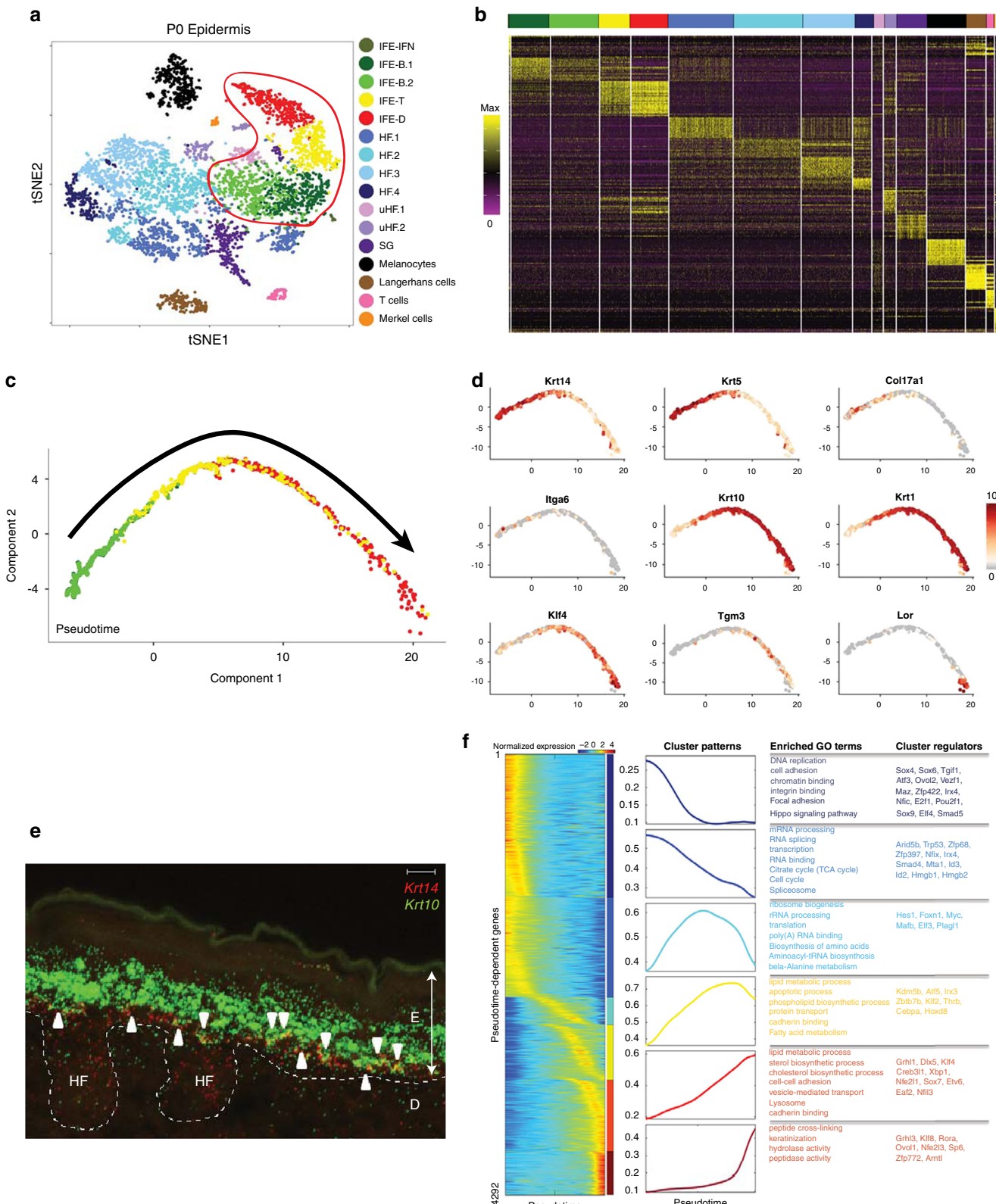

genes that are dynamically expressed over the pseudotime trajectory. K-medoid clustering grouped these genes into six gene modules with distinct expression patterns and biological functions (Fig. 1f)[9,10,14]. Module 1, high early and falls sharply, is enriched in genes associated with cell proliferation. Module 2, also high early but falls more slowly, is enriched in genes involved in transcription and mRNA processing. Module 3,

peaks in the middle of the trajectory, is enriched in genes for ribosome biogenesis and protein translation. Module 4, peaks towards the end, is enriched in lipid biosynthesis genes. Module 5, also peaks toward the end but with a slower rise, is also enriched in lipid synthesis pathways and in cell adhesion genes. Module 6, a sharp peak at the end, is enriched in crosslinking and barrier genes.

**Fig. 1 IFE differentiation is gradualistic and features numerous transition cells. a** tSNE plot showing all 16 epidermal subpopulations of the WT P0 mouse back epidermis; IFE cells are in the red-outlined box. HF hair follicle, uHF upper hair follicle, SG sebaceous gland. **b** Heatmap showing the expression of top 20 marker genes for all 16 epidermal subpopulations. Gene lists in Supplementary Data 1. **c** Pseudotime analysis of the IFE with cluster identity from **a** projected on the trajectory. The trajectory goes from basal to transition to differentiated cells without major branches (arrow), consistent with unidirectional IFE differentiation. **d** Expression of canonical markers for distinct stages of epidermal differentiation. **e** RNA-FISH for *Krt10* and *Krt14* in WT P0 mouse epidermis. White arrowheads point to yellow *Krt10*/*Krt14* double-positive cells. E epidermis, D dermis, HF hair follicle. Scale bar = 10 μm, representative image for $N = 5$. **f** Expression heatmap and expression pattern of 4,292 pseudotime-dependent genes that cluster into six gene modules. Also shown are the gene ontology and example transcription factors for each gene module. Gene lists in Supplementary Data 1.

Each module contains transcription factors with known functions in IFE differentiation that match the overall gene functional category of that cluster. At early differentiation stages, transcription factors associated with the cell cycle and DNA repair are highly expressed, including *Trp53*, *Smad4*, *Hmgb1*, and *Hmgb2*. As differentiation commences, *Mafb* and *Myc* become highly expressed whereas late differentiation transcription factors such as *Klf4* and *Grhl3* become highly expressed at late and terminal differentiation stages. In addition, we have identified many candidate transcription factors with heretofore unknown roles in the IFE (Fig. 1f).

Although these findings correspond well with known features of IFE differentiation, they also point to an unusual relationship between cell proliferation and protein synthesis which are normally tightly coupled; in the IFE, peak gene expression for protein synthesis appears after peak expression of cell cycle genes. Furthermore, these data indicate that the majority of IFE differentiation genes exhibit expression that is not layer-specific with at least four out of six gene modules straddling different layers. This result and the observed high number of transition cells challenge the traditional model of stepwise IFE differentiation, suggesting that at a transcript level, IFE differentiation is continuous and gradualistic.

**The IFE basal layer contains two distinct cell populations**. Next we sought to better decipher P0 basal IFE stem cell heterogeneity by examining the unique molecular signatures expressed in the two basal subpopulations (Figs. 1a and 2a). IFE.B1 cells (446 cells) express *H19*, *Igf2*, and *Wnt4* at a higher level than IFE.B2 cells, whereas IFE.B2 cells (556 cells) exclusively express *Sox4* and *Fst*, as well as other markers, including *Dcn* (Fig. 2a, b). We performed principle component analysis (PCA) on these two populations and found the above markers to be among the genes that explain most of the variances in the first PC (Supplementary Fig. 2A); in contrast, cell cycle genes do not contribute to variances in the first few PCs. To confirm that active cell cycling does not play a major role in defining the two basal clusters, we also regressed out the cell cycle genes and reclustered the basal IFE cells, which did not alter the results (Supplementary Fig. 2B–D). Therefore, we ruled out the possibility that these clusters are due to differences in cell cycle stages among the cells. Furthermore, IFE.B1 and B2 are not distinct stages of differentiation on the pseudotime trajectory (Supplementary Fig. 2E). In addition, we used an independent clustering method, SC3[15], to recluster all basal IFE cells; SC3 clustering results were very similar to Seurat's, confirming the robustness of this finding (Fig. 2c).

Our data, then, highlight IFE basal cell heterogeneity, suggesting there are two basal stem cell populations in the P0 IFE. To validate our findings, we carried out RNA-FISH studies with *H19* and *Wnt4* probes. We found that *Wnt4*-high and *Wnt4*-low cells are distributed heterogeneously within the IFE basal layer (Fig. 2d) and that high *H19* expression tends to co-localize with the *Wnt4*-high cells (Supplementary Fig. 2F). These findings indicate that the IFE.B1 and IFE.B2 cells intermingle in the basal layer and are not spatially distinct groups of cells. Our

findings are in agreement with previous studies showing that Wnt-secreting stem cells play a central role in IFE self-renewal during homeostasis[16–18].

**The IFE differentiation program is established at E14.5**. To investigate when the above-described characteristics of the IFE appear and how the differentiation program may change during development, we generated single cell suspensions from littermate WT and *Grhl3*$^{-/-}$ mouse dorsal whole skin (E14.5, E16.5) and epidermis (E18.5/P0). We then performed scRNA-seq on two biological replicates from each time point for each genotype (Supplementary Figs. 3A–F, 4A–F, and 5A, B). In our computational analysis, we focused first on normal IFE development, selecting IFE cells from all three time points (Fig. 4a, Supplementary Fig. 3A) and clustering them jointly into six cell states (Fig. 3a–c) based on canonical markers of IFE differentiation (Fig. 3c). We observed basal, transition, spinous, and terminally differentiated IFE clusters at each of the three developmental time points, indicating that the full IFE differentiation program is functional as early as E14.5 (Fig. 3a).

Even though we observed the full range of IFE differentiation states across the developmental time points, there are changes in the proportion of cell states when comparing the E14.5 with the P0 IFEs (Fig. 3b). The proportion of basal cells decreases as the skin develops (IFE-B; 52% at E14.5, 58% at E16.5, 37% at P0). In contrast, the proportions of transition cells (IFE-T; 19% at E14.5, 19% at E16.5, 23% at P0), differentiated cells (IFE-D; 22% at E14.5, 15% at E16.5, 27% at P0), and terminally differentiated cells (IFE-TD; 7% at E14.5, 8% at E16.5, 13% at P0) all increase. The E14.5 IFE.T cells, however, are different from IFE.T cells of the P0 IFE in that the E14.5 IFE.T cells are highly proliferative (Fig. 3c and Supplementary Fig. 3G, H), consistent with known features of the intermediate layer at this developmental stage. We observed this feature to a lesser extent for the E16.5 IFE.T cells (Fig. 3c and Supplementary Fig. 4G, H). The pseudotemporal ordering of E14.5 and E16.5 IFE cells separately shows linear trajectories with a single branch consisting mostly of proliferating IFE.T cells (Supplementary Figs. 3G and 4G). This finding corroborates the idea that IFE cells goes through a state of rapid proliferation before differentiating at E14.5, a feature that decreases at E16.5, and is abolished at P0.

scEpath pseudotemporal trajectory analysis[14] on the integrated data from all three developmental time points revealed a linear differentiation trajectory with a small outlying cluster mainly composed of proliferative IFE.T cells from E14.5 and E16.5 (Fig. 3d, e). Consistent with our clustering results, the distal end of the trajectory containing IFE.TD cells extends further in the P0 than the E14.5 and the E16.5 IFEs (Fig. 3d). Across this integrated trajectory, 2801 genes show pseudotime-dependent change in expression; these can be grouped into six gene modules, which contain 149 transcription factors (Fig. 3f). To investigate how gene expression programs change during IFE development, we compared the differentially expressed genes in IFE cells across the three time points. We found that in IFE cells collectively, E14.5 and E16.5 cells display higher expression of genes for growth,

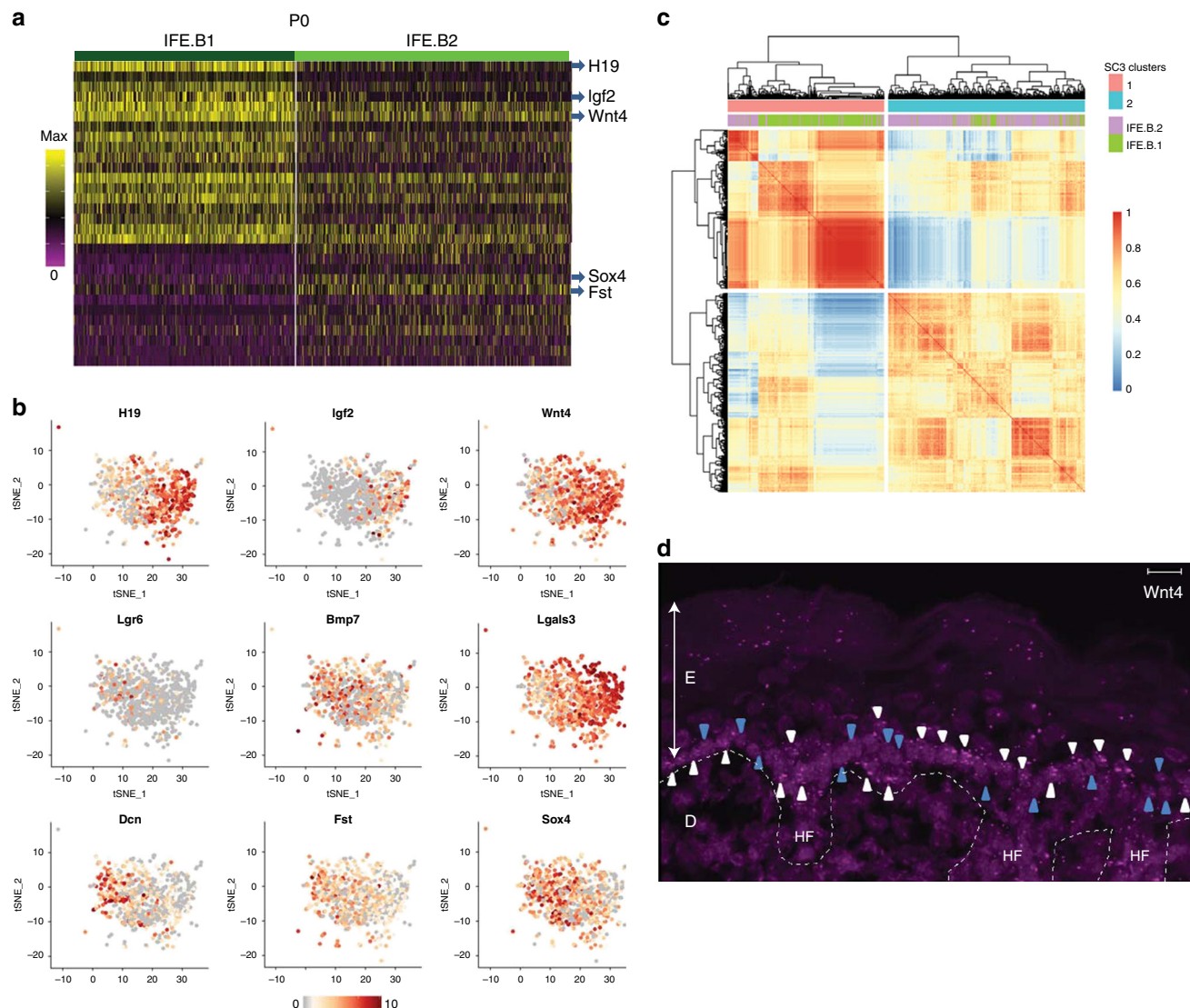

**Fig. 2 The P0 IFE basal cell layer contains two distinct cell populations. a** Expression heatmap of unique gene signatures for each of the two basal IFE clusters. Blue arrows highlight select differentially expressed genes between the two basal clusters. Gene lists in Supplementary Data 1. **b** Expression level of select genes, which are enriched in either IFE.B1 or IFE.B2, projected onto tSNE for the two basal clusters. **c** Heatmap showing the correlation between cells for SC3 clustering (1, 2; red, blue) and Seurat clustering (IFE.B1, IFE.B2; purple, green). **d** RNA-FISH for *Wnt4* in the P0 IFE. White arrowheads, *Wnt4*-high cells; blue arrowheads, *Wnt4*-low cells; white broken line, basal lamina; E epidermis, D dermis, HF hair follicle. Scale bar = 10 μm, representative image for *N* = 5.

nucleic acid metabolism, and RNA regulation, whereas P0 cells display higher expression of genes for cell–cell adhesion and barrier formation (Fig. 3g). In IFE.B cells selectively, there is enrichment for cell cycle and cell migration at E16.5, whereas there is greater enrichment for nuclear division and cell growth at P0 (Fig. 3h). In IFE.D and IFE.TD cells, there is enrichment for myeloid cell differentiation, cytokine secretion and ERK signaling at E16.5, whereas there is enrichment for lipid metabolism and morphogenesis at P0 (Fig. 3i). Taken together, these data indicate that the full range of the IFE differentiation program is already established as early as E14.5, and that epidermal development from this time point is characterized by a subtle shift toward differentiated cell states and away from growth and proliferation.

**GRHL3 is required for tempering IFE stem cell number.** Previous work on GRHL3 suggested that all layers of the IFE do develop in *Grhl3*−/− embryos, although at P0 the basal layer is disorganized and the spinous and granular layers appear thicker

than normal[7,8]. Bulk gene expression measurement and other studies suggested that GRHL3 transcriptionally activates the terminal differentiation gene expression program in the granular layer[7,8]. The differentiation-specific function of GRHL3 is also consistent with the restricted expression of GRHL3 in the most differentiated layers of the IFE at P0. How GRHL3 affects IFE cell states has not been investigated.

Hence, we tested how GRHL3 affects P0 IFE differentiation as detected by scRNA-seq. We performed scRNA-seq on single cell suspensions from the back-skin epidermis of two WT and two *Grhl3*−/− littermates at P0, capturing a total of 29,796 cells after quality control (WT1 = 6766, WT2 = 6590, *Grhl3*−/−1 = 9157, *Grhl3*−/−2 = 7283) (Supplementary Fig. 5A, B). As expected, we recaptured the IFE cell heterogeneity depicted in Fig. 1 (Fig. 4a, b), including the basal keratinocyte heterogeneity, thus validating these earlier findings (Figs. 1 and 2). But here, we identified a third, actively proliferating basal population (IFE.B3) (Fig. 4a and Supplementary Fig. 5c); this cluster most likely emerged due to

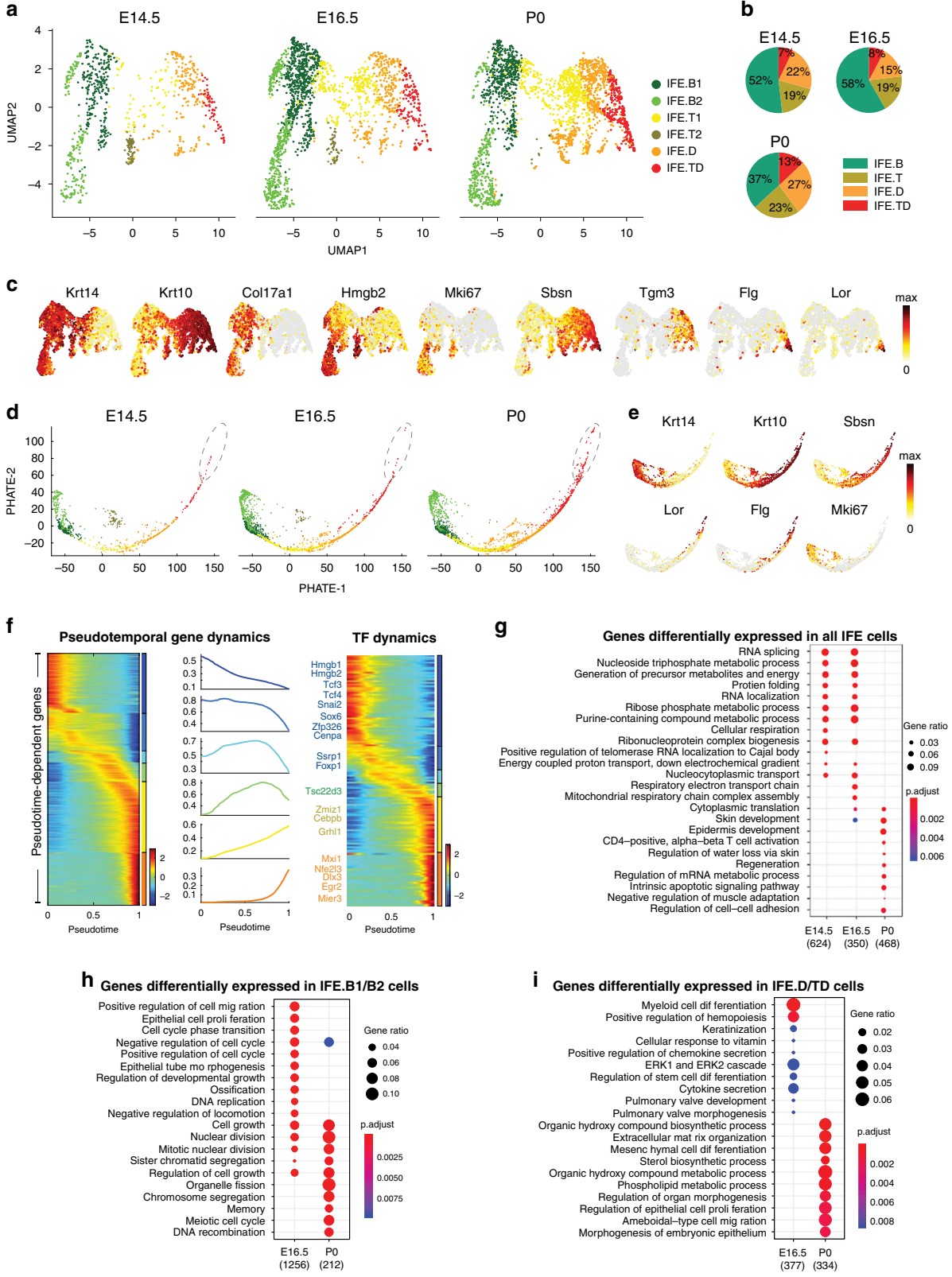

increased total number of cells captured across the four samples compared to the experiment in Figs. 1 and 2. We also believe that the higher number of cells included in this experiment allowed the terminally differentiated (IFE.TD) IFE cells to form a distinct cluster.

We integrated all four samples[19] and found that although the proportion of each of the IFE subpopulations is consistent

between the two wild type epidermis, the cell composition of the mutant epidermis is clearly different from the wild type (Fig. 4b). The terminally differentiated IFE population (IFE.TD) is almost absent in the $Grhl3^{-/-}$ IFE (Fig. 4a, b), suggesting that in addition to activating the expression of terminal differentiation genes, GRHL3 promotes the formation of the most differentiated cells of the IFE. Most strikingly, however, the proportion of basal cells

**Fig. 3 The IFE Differentiation program is already established at E14.5. a** UMAP showing distribution of cell states as defined in the P0 IFE (Fig. 1) in the developing IFE (E14.5, E16.5, and P0). We observe two transition populations, IFEE.T1 and IFE.T2, at E14.5 and E16.5. IFE.T2 cells are proliferating transition cells. **b** Pie charts showing the proportion of major cell states at each developmental time point. IFE.B and IFE.TD decrease and increase, respectively, across development. **c** The expression of the indicated differentiation and proliferation markers projected onto the combined UMAP for all three developmental time points in **a**. **d** PHATE maps of cell populations defined in **a**, showing pseudotime progression at each developmental stage. The gray circle emphasizes the extension of terminal differentiation at P0. **e** Expression of the indicated differentiation and proliferation markers onto the combined PHATE maps from **d**. **f** Left panel, expression heatmap and expression pattern of 2801 pseudotime-dependent genes from the combined PHATE maps from **d**; these cluster into six gene modules. Right panel, expression heatmap of 149 pseudotime-dependent transcription factor (TF) genes from the combined PHATE maps from **c**. Representative TFs from each gene modules are color-coded and indicated on the left. Gene lists in Supplementary Data 1. **g** Gene ontology categories of genes differentially expressed at each developmental stage in all IFE cells. Gene lists in Supplementary Data 1. **h** Gene ontology categories of genes differentially expressed at E16.5 and P0 in IFE.B1 and IFE.B2 cells. Gene lists in Supplementary Data 1. **i** Gene ontology categories of genes differentially expressed at E16.5 and P0 in IFE.D and IFE.TD cells. Gene lists in Supplementary Data 1.

(IFE.B1, IFE.B2, and IFE.B3) increased, whereas the proportion of transitioning (IFE.T) and differentiated cells (IFE.D) decreased (Fig. 4b and Supplementary Fig. 5D) in the $Grhl3^{-/-}$ IFE. In addition, we detected a substantial number of cells representing IFE states that are nearly exclusive to the $Grhl3^{-/-}$ epidermis. We termed these two populations aberrant IFE 1 and 2 (IFE.A1, IFE.A2). About 100 IFE.A1 cells were found in each of the two $Grhl3^{-/-}$ biological replicates, whereas a greater number (135) of IFE.A2 cells were found in $Grhl3^{-/-}$ 2 than $Grhl3^{-/-}$ 1 (18) (Fig. 4a, b). We observed no clear changes in cellular composition between WT and $Grhl3^{-/-}$ IFE cells at E14.5 and E16.5, suggesting that the cell state differences in the $Grhl3^{-/-}$ IFE arose between E16.5 and P0 (Supplementary Fig. 5E). These findings indicate that in addition to promoting terminal differentiation, a major function of GRHL3 is to suppress the abundance of epidermal stem cells and the formation of abnormal IFE cell states at the later stages of IFE development.

**GRHL3 suppresses aberrantly differentiated IFE progenitors**. The IFE.A1 cells resemble the normal transitioning cell state (IFE.T) in that they express both basal and suprabasal IFE gene signatures (Supplementary Fig. 6A). Consistently, we find IFE.A1 cells between the basal and differentiated IFE cells on PCA analysis (Supplementary Fig. 6B). RNA-FISH staining of the IFE showed an increase in the number of cells in the basal and spinous layers expressing both basal (*Krt14*) and suprabasal (*Krt10*) markers in the $Grhl3^{-/-}$ epidermis compared to WT (Fig. 4d). Immunofluorescence staining corroborates this at the protein level (Supplementary Fig. 6C), suggesting that the IFE.A1 cells are located at the basal-spinous boundary. Yet, the IFE.A1 population also expresses genes distinct from wild type IFE.T cells, including *Sprr2a3*, *Tpm2*, and *Ly6a* (Fig. 4c, Supplementary Fig. 6D). Furthermore, cell cycle analysis[19,20] on all IFE subpopulations revealed that IFE.A1 cells are more proliferative than the IFE.T cells; in fact, a subset of IFE.A1 cells express proliferation genes to a similar level as the dividing basal cluster IFE.B3 (Supplementary Fig. 6E, F). These findings indicate that GRHL3 suppresses the formation of aberrantly differentiated epidermal progenitors. Given the unexpected change in basal cell composition and the emergence of aberrant transition-like cells in the $Grhl3^{-/-}$ IFE, we conclude that GRHL3 is required for maintaining proper IFE cell composition, acting as early as the transition state between basal and spinous cells.

**GRHL3 loss disrupts differentiation at the IFE.B–IFE.T transition**. To further define the differentiation stage in which GRHL3 functions during IFE differentiation, we constructed separate pseudotime trajectories for WT and $Grhl3^{-/-}$ IFE cells. In agreement with our earlier experiment (Fig. 1c), the two WT IFE samples formed a linear trajectory going from basal to terminally differentiated cells (Fig. 5a, b and Supplementary

Fig. 7A). In the $Grhl3^{-/-}$ IFE, the aforementioned expansion of proliferating basal cells and the presence of the IFE.A1 subpopulation both contributed to a branching of the linear differentiation trajectory (Fig. 5a). The branching occurred at the junction between basal cells and IFE.T cells. The tip of the aberrant branch in the $Grhl3^{-/-}$ IFE is composed of highly proliferating basal IFE cells (IFE.B3) and the IFE.A1 cells. Consistent with the Seurat clustering (Fig. 4a) and the RNA-FISH experiments (Fig. 4d), this pseudotime analysis also suggests that the IFE.A1 cell state is most similar to that of IFE.T cells. In contrast, the IFE.A2 cells are more similar to cells at a later differentiated stage, albeit with higher proliferation (Supplementary Fig. 6E, F). The IFE.A2 cells also express mitochondrial genes and other markers of cellular stress (Supplementary Data 1), which may explain the variability in this cell population between the two replicates; we speculate that the IFE.A2 population may be a sensitive marker of the cellular damage associated with hyperproliferation and defective barrier. In sum, besides being a transcriptional activator of late differentiation genes, GRHL3 performs important functions at early stages of IFE differentiation.

**RNA-velocity reveals a differentiation commitment point**. To further understand the differentiation abnormalities in the $Grhl3^{-/-}$ IFE, we applied RNA-velocity, a method that exploits the relative abundance of nascent (unspliced) and mature (spliced) transcripts to predict the future state of individual cells on a time scale of hours[21,22]. This method can infer directionality and dynamics of small groups of cells with respect to each other in the pseudotime differentiation trajectory. In these data, the direction of the arrows point to the fate the cells are heading toward whereas the length of the arrows reflects how fast the cells are heading in that direction.

We expected that RNA-velocity would reveal arrows with a uniform direction from IFE.B to IFE.TD in the pseudotime trajectory of the WT IFE. Surprisingly, we found that during WT IFE differentiation, the basal and transition cells exist in a dynamic equilibrium state in which no clear directionality is observed among them (Fig. 5c). Once the cells reach the IFE.T/ IFE.D boundary, the arrows become longer and align in the direction of terminal differentiation in a uniform manner, suggesting that after cells exit the IFE.T state, they commit to differentiate in a uniform manner until they terminally differentiate (Fig. 5c). These results are not exclusive to this particular dimension reduction method as we found similar results with PCA of all IFE cells, which also happens to delineate IFE's linear differentiation trajectory from IFE.B to IFE.T to IFE.D (Supplementary Fig. 7B). In the $Grhl3^{-/-}$ IFE, much fewer cells pass the IFE.T/IFE.D commitment point where unidirectional IFE differentiation commences. Strikingly, early transition cells display a clear propensity back toward the basal cell fate, again

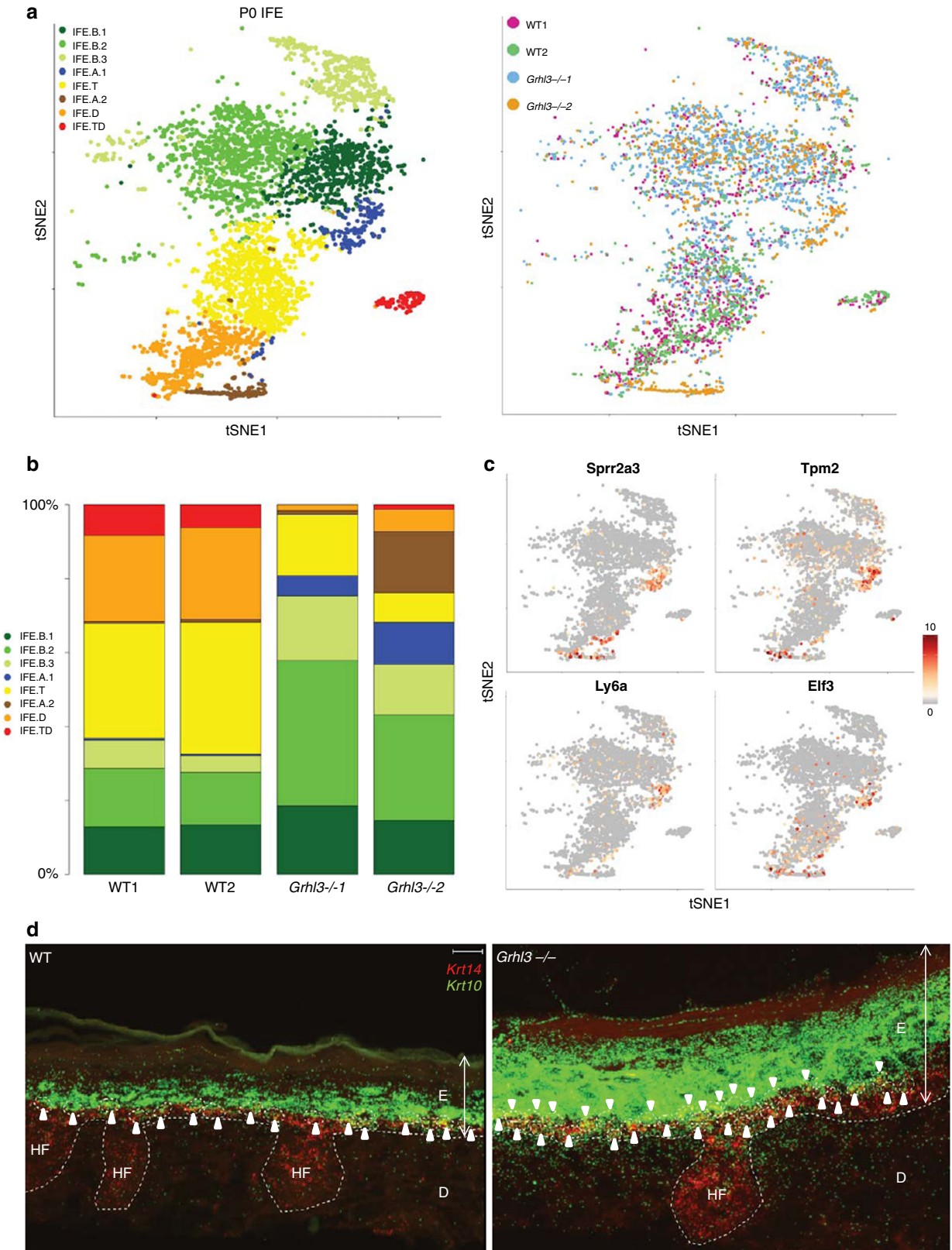

**Fig. 4 GRHL3 is required for suppressing aberrant basal IFE stem cells. a** tSNE plot showing WT and *Grhl3*^{−/−} P0 IFE cells colored by clusters (left) and genotype (right). **b** Bar plot showing the fraction of each IFE cell type in each sample. There is expansion of basal cells and reduction of differentiated/terminally differentiated cells in *Grhl3*^{−/−} IFE. **c** Gene expression levels of markers for the IFE.A1 population. *Sprr2a3, Tpm2, Ly6a* and *Eif3* are shown on a tSNE plot of the P0 IFE cells. **d** Representative RNA-FISH of *Krt14* and *Krt10* transcripts in P0 WT and *Grhl3*^{−/−} mouse epidermis; white arrows highlight *Krt14/Krt10* double-positive transition cells. Note the higher number of double-positive cells in the *Grhl3*^{−/−} epidermis; also some transition cells are found in the spinous layer. N = 5. White broken line, basal lamina; E epidermis, D dermis, HF hair follicle. Scale bar = 10 μm.

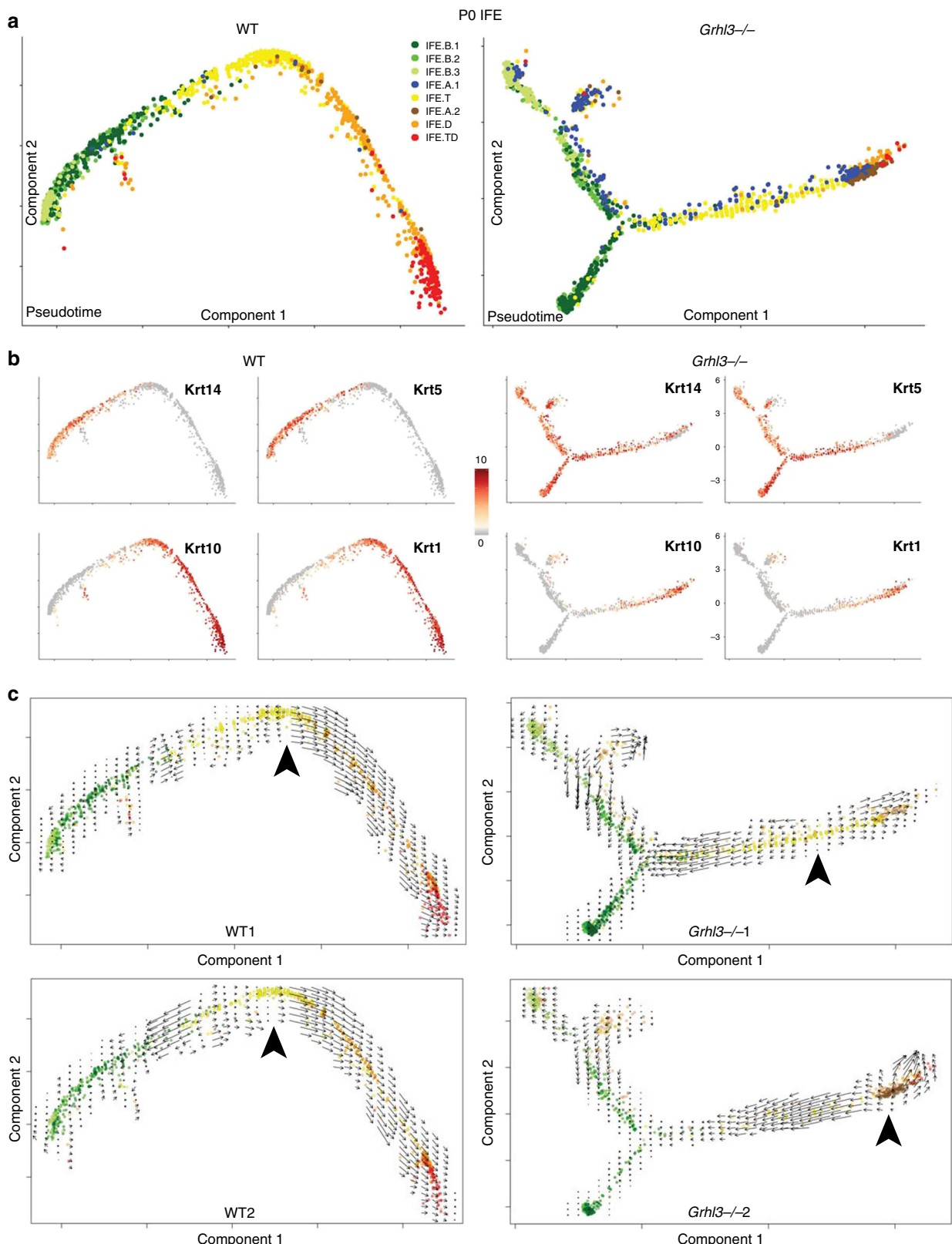

suggesting that GRHL3 plays an important role in promoting the differentiation of IFE.B to IFE.T cells.

These findings suggest: (1) Normal IFE differentiation proceeds in a smooth, continuous manner rather than in punctuated stages. (2) There are crucial changes in the regulation of IFE differentiation at the IFE.T/IFE.D boundary, where cells commit to differentiate and at which GRHL3 plays a key role. (3) Early progenitor cells transitioning to differentiation are highly plastic in their progenitor-differentiation fate decisions. (4) There is aberrant signaling in the $Grhl3^{-/-}$ epidermis that actively drives early transition cells toward the basal fate.

**Fig. 5 RNA velocity reveals a commitment point at the IFE.T/IFE.D transition. a** Pseudotime trajectories showing IFE differentiation going from IFE-B to IFE-T to IFE-D to IFE-TD for the two WT (left) and two *Grhl3*$^{-/-}$ (right) P0 IFEs. The *Grhl3*$^{-/-}$ differentiation trajectory has a branch which is attributed to the IFE-A1 population and expansion of basal, proliferating cells. **b** Expression of *Krt14*, *Krt5*, *Krt10*, and *Krt1* projected on the pseudotime trajectory for the WT (left) and the *Grhl3*$^{-/-}$ (right) IFE. **c** RNA-Velocity analysis of the pseudotime trajectory of IFE cells for WT1, WT2, *Grhl3*$^{-/-}$1, *Grhl3*$^{-/-}$2. Direction of the arrows points to the fate the cells are heading toward; length of the arrows reflects how fast the cells are heading toward a particular fate. Note IFE differentiation is marked by a clear commitment point (arrowhead) and is a one-step, continuous process with no intermediary stages in the WT. In contrast, most cells fail to proceed through the commitment point (arrowhead) and transition cells show a propensity to go back to basal fate in the *Grhl3*$^{-/-}$ IFE.

**GRHL3 represses Wnt signaling in the IFE.** In the P0 *Grhl3*$^{-/-}$ IFE, the basal compartment is expanded (Fig. 4b) and IFE.T cells exhibit a propensity to go backward toward the basal fate as shown with RNA velocity analysis (Fig. 5c). These findings suggest that GRHL3 is required for suppressing signaling pathways that actively drive IFE cells toward the basal fate. Previous mouse experiments showed that Wnt-secreting keratinocyte stem cells are distributed in the basal IFE and that autocrine Wnt signaling is required for self-renewal in mouse adult epidermal homeostasis[16,18]. Hence, we hypothesized that aberrant Wnt signaling could explain the change in cell composition and disruption of differentiation in the *Grhl3*$^{-/-}$ IFE.

To determine if Wnt signaling is altered in the *Grhl3*$^{-/-}$ IFE, we compared the expression level of Wnt signaling pathway components between P0 WT and *Grhl3*$^{-/-}$ IFE populations in our scRNA-seq data. We observed increased expression of genes such as *Wnt4*, *Lef1*, *Ctnnb1*, *Gpc3*, and *Gsk3b* in the *Grhl3*$^{-/-}$ basal IFE populations and the IFE.A (Fig. 6a and Supplementary Fig. 8A). As *Grhl3* is expressed at highest levels in the spinous and granular layers in WT mice, we also considered that GRHL3 could suppress Wnt signaling in a paracrine manner by regulating Wnt inhibitory ligand expression. Consistent with this idea, we found that *Dkkl1*, encoding a known Wnt pathway antagonist[23], is expressed at a high level in the spinous and granular layers but not in the basal layer (Fig. 6b). In the *Grhl3*$^{-/-}$ IFE, *Dkkl1* expression level is reduced, as are the number of cells expressing *Dkkl1* (Fig. 6b and Supplementary Fig. 8C). To validate these findings, we performed RNA-FISH for the Wnt ligand *Wnt4* and the Wnt downstream effector *Lef1* in WT and *Grhl3*$^{-/-}$ IFE; both *Wnt4* and *Lef1* are expressed at a higher level in the *Grhl3*$^{-/-}$ IFE (Fig. 6c). Furthermore, whereas *Wnt4* and *Lef1* expression is restricted to a subset of basal cells in the WT IFE, they are expressed more widely in basal cells and in cells of the spinous layer in the *Grhl3*$^{-/-}$ IFE (Fig. 6c). In RNA-FISH experiments, we also found that *Dkkl1* is expressed at a lower level in the suprabasal compartment of the *Grhl3*$^{-/-}$ IFE (Fig. 6d). These RNA-FISH results are consistent with the scRNA-seq results showing an increase in the number of keratinocyte stem cells (IFE.B) in the *Grhl3*$^{-/-}$ IFE and the fact that the IFE.A population, which is enriched in the *Grhl3*$^{-/-}$ IFE, is a transition population expressing high level of *Wnt4*.

Noting a connection between GRHL3 and Wnt signaling, we explored our previously published GRHL3 ChIP-seq data from the E16.5 epidermis[24]. We found that GRHL3 directly binds to the promoters of key Wnt signaling components, including *Wnt4*, *Lef1*, *Ctnnb1*, *Gsk3b* and *Ctnnbip1* (Fig. 7a and Supplementary Fig. 8B), and near gene bodies of *Axin2*, *Myc* (Supplementary Fig. 8B). These data are consistent with GRHL3 directly regulating Wnt signaling to maintain proper balance between proliferation and differentiation in the IFE. Next, we set out to determine if suppressing Wnt signaling can decrease the thickness of the hyperthickened IFE in *Grhl3*$^{-/-}$ mice at P0. We harvested P0 mouse back skin and used Wnt-C59, a small molecule antagonist of the Wnt pathway, or DMSO to treat samples from the same skin ex vivo for 48 h in media. Wnt-C59

effectively suppressed Wnt signaling in the epidermis (Fig. 7b). Meanwhile, the Wnt antagonist significantly decreased the thickness of the hyperplastic *Grhl3*$^{-/-}$ IFE (Fig. 7c, d). Taken together, these results reveal a previously unexpected role of GRHL3 in suppressing Wnt signaling, thereby promoting the transition from basal to transition IFE cells.

## Discussion

Here, we report single cell transcriptome profiling study of the developing mouse epidermis at E14.5, E16.5, and P0 in WT and *Grhl3*$^{-/-}$ mice. The data includes a total of 85,286 cells, a valuable resource available for analysis beyond the scope of this paper. We identified all known cell types of the epidermis and defined unique gene signatures for each cell type. We have also uncovered numerous hitherto unknown genes important for IFE differentiation and development from E14.5 to P0.

Our study, which focused on IFE differentiation and its regulation by GRHL3, has revealed a number of insights: (1) At a transcript level, IFE differentiation is continuous and gradualistic rather than stepwise (Fig. 1); (2) There is a large number of basal-spinous transition cells (Fig. 1); (3) An important regulatory change occurs at the IFE.T/IFE.D boundary where IFE cells commit from a plastic state to an irreversible fate toward terminal differentiation (Fig. 5c); (4) There are two main populations of basal stem cells and a third basal cell state linked to cell proliferation (Figs. 1, 2, and 4); (5) The full IFE differentiation program is already operating as early as E14.5 (Fig. 3); (6) In addition to a role in promoting terminal differentiation in the most differentiated cells, GRHL3 is required at an earlier differentiation stage to suppress Wnt signaling and to suppress the expansion of stem cells (Figs. 4, 6, 7).

Clearly, the IFE contains morphologically distinct layers. Yet, our scRNA-seq findings challenge the conventional notion of stepwise IFE differentiation (Supplementary Fig. 8D). At a transcript level, differentiation occurs in a continuous manner from the most primitive stem cell to the most terminally differentiated IFE cell at the top of the epidermis. The demonstration of a large number of transition cells also supports the gradualistic model for IFE differentiation. Consistent with previous results, our findings suggest that basal stem cells (IFE.B) exist in a reversible equilibrium with a large population of transition cells (IFE.T) and that is not until after passing the transition cell stage that IFE cells fully commit to differentiation. In the future, live imaging likely will be a powerful tool to investigate the gradualistic feature of IFE differentiation with quantitative, temporal resolution.

Studies on stem cells in the basal layer of the adult epidermis have suggested cell heterogeneity within this layer, although the nature of this heterogeneity remains controversial. The hierarchical model of stem cell differentiation suggests that the basal layer comprises rare slow-cycling long-term stem cells and their fast-cycling committed progenitors[25,26]. Another related model suggests that slow-cycling and fast-cycling stem cells occupy distinct regions of the basal layer and renew within their respective regions[27,28]. By contrast, the stochastic model of stem cell differentiation suggests that all basal cells are equivalent in

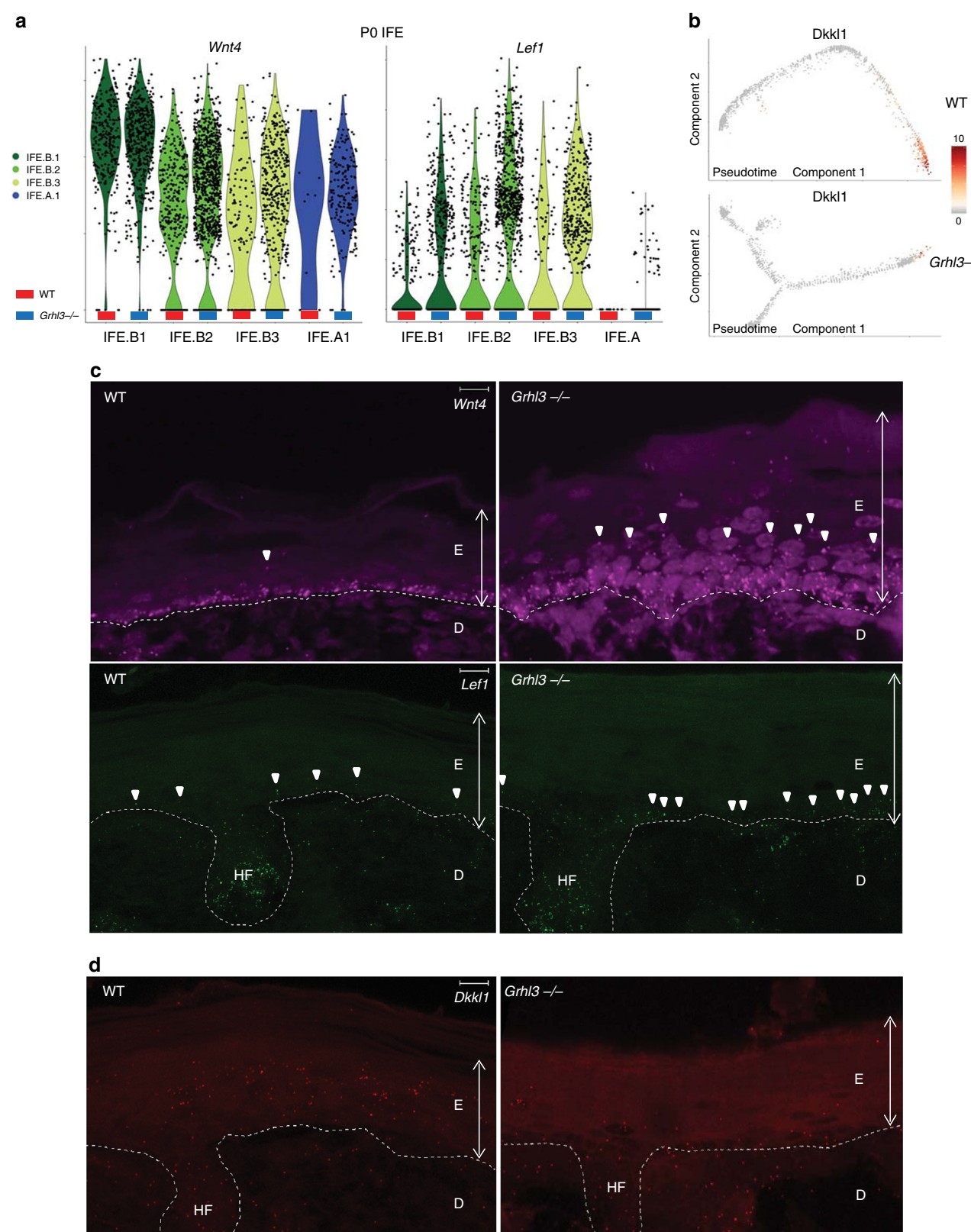

terms of their stemness and that they differentiate in a stochastic manner[16,29–31]. We found two populations of stem cells in the P0 IFE with one major difference between these being the higher expression of *Wnt4* in IFE.B1. Wnt4 was previously shown to be important for IFE stem cell self-renewal[16] and our RNA-FISH results also support a model in which *Wnt4*-high stem cells are distributed randomly throughout the basal IFE. We propose that *Wnt4*-high basal IFE cells are workhorses of IFE maintenance as increased *Wnt4* due to *Grhl3* loss leads to a considerable change in cell composition of the IFE.

Previous work on GRHL3 has focused on suprabasal IFE cells where *Grhl3* expression is the highest and where it regulates

**Fig. 6 GRHL3 represses Wnt pathway gene expression in basal IFE cells. a** Violin plot showing the *Wnt4* (left) and *Lef1* (right) expression for each cell among the basal cell types in the WT (both biological replicates) and the *Grhl3*$^{-/-}$ (both biological replicates) IFE. There is higher number of *Wnt4*-high and *Lef1*-high cells in all basal and aberrant populations in the *Grhl3*$^{-/-}$ than in the WT. **b** Expression of *Dkkl1* projected on the pseudotime trajectory for the WT (top) and the *Grhl3*$^{-/-}$ (bottom) IFE. Note the decreased number of cells expressing *Dkkl1* in the *Grhl3*$^{-/-}$ IFE. **c** Representative RNA-FISH images of *Wnt4* (upper panels) and *Lef1* (lower panels) in WT and *Grhl3*$^{-/-}$ P0 epidermis. There is increased *Wnt4* expression and aberrant expression of *Wnt4* in cells above the basal layer in *Grhl3*$^{-/-}$ IFE as indicated with white arrowheads. There is increased *Lef1* expression in the basal layer in the *Grhl3*$^{-/-}$ IFE as indicated with white arrowheads. N = 3 WT and *Grhl3*$^{-/-}$ littermates. E epidermis, D dermis, HF hair follicle. **d** Representative RNA-FISH image of *Dkkl1* in WT and *Grhl3*$^{-/-}$ P0 epidermis. There is decreased *Dkkl1* expression in the suprabasal layers in *Grhl3*$^{-/-}$ IFE. N = 3 WT and *Grhl3*$^{-/-}$ littermates. White broken lines trace the basal lamina. E epidermis, D dermis, HF hair follicle. Scale bars = 10 μm.

terminal differentiation genes[7,8]. scRNA-seq allowed us to quantitatively characterize the cell compositional change in *Grhl3*$^{-/-}$ epidermis and uncover another important function of GRHL3, which is to suppress expansion of IFE stem cells and Wnt signaling in the basal layer. RNA velocity analysis suggests that the intermediate cell state (IFE.T) is plastic and can either revert to the basal stem cell (IFE.B) state or progress toward differentiation (IFE.D). Unexpectedly, we find that GRHL3 promotes the progression of the IFE.B state to the IFE.T state, which accounts for the prominent accumulation of epidermal stem cells in the *Grhl3*$^{-/-}$ IFE. Based on our findings of increased Wnt signaling activity in the *Grhl3*$^{-/-}$ IFE and on previous work showing that Wnt signaling promotes IFE stem cell proliferation and maintenance[16,17], we postulate that an important developmental role of GRHL3 is to temper Wnt signaling in IFE stem cells.

As GRHL3 binds to the promoters of several Wnt signaling genes, including *Wnt4* and *Lef1*, this effect may be from direct transcriptional repression of key Wnt effectors; we have previously shown that GRHL3 binding to promoters is often associated with transcriptional repression[32]. This model is consistent with data from human keratinocytes suggesting that GRHL3 may act early in maintaining proper proliferation/differentiation balance[33,34]. But our scRNA-seq data also suggests that paracrine mechanisms could be at play. The secreted Wnt pathway antagonist DKKL1, which is normally expressed in the terminally differentiated cells of the IFE, is decreased in the *Grhl3*$^{-/-}$ IFE, thereby potentially decreasing the suppression of Wnt signaling in the basal layer.

## Methods

**Mouse work**. One male and one female for each genotype (WT (C57Bl/6J), *Grhl3*$^{-/-}$ mice[7]) for each time point were used. Timed pregnancies were used to obtain embryos for E14.5, E16.5, and E18.5; the developmental stage of embryos was verified using external features of the embryos[35]. All experiments were performed on WT and *Grhl3*$^{-/-}$ littermates from the same pregnancy. Mice were fed food and water ad libitum and maintained on a regular 12 h day/night cycle. All animal experiments were performed in accordance with Institutional Animal Care and Use Committee at University of California, Irvine (Protocol No. AUP-19-012).

**Tissue isolation**. All *Grhl3*$^{-/-}$ embryos displayed spina bifida. For E18.5/P0 mice, we removed approximately 6 mm radius circular sample of backskin, centered at the midline in one axis and on a line drawn between the forelimbs in the other axis. The same region of skin was microdissected for E14.5 and E16.5 embryos. This anterior location avoids skin near the spina bifida defect. For E14.5 and E16.5, the micro-dissected embryonic back skin was incubated in 2.5 U/ml Dispase (STEM-CELL Technologies) in EpiLife medium (Thermo Fisher Scientific) for 2 h at room temperature (RT). The tissue were then washed with media and incubated in Accutase (STEMCELL Technologies) for 30 min at RT followed by dissociation into single cells. For P0, the back skin was incubated in 2.5 U/ml Dispase in EpiLife medium overnight. The epidermis was then manually separated from the dermis and incubated in Accutase for 30 min at RT followed by dissociation into single cells. For all time points, the dissociated cell suspension was strained with 40 μM filter and washed with media. Dead Cell Removal kit (Miltenyi Biotec) was used to remove dead cells prior to resuspension in 0.04% BSA (Thermo Fisher Scientific). Chromium Single Cell 3′ v1/v2 (10x Genomics) library preparation was then performed by the University of California, Irvine, Genomic High Throughput Facility (UCI-GHTF) according to manufacturer's protocol.

**Immunofluorescence localization of markers**. Fresh frozen OCT 10 μM sections were incubated in acetone at −20 °C for 10 min, washed with TBS, fixed in 4% PFA for 10 min, and then washed with TBS three times. Tissues were then permeabilized using TBS with 0.3% TritonX-100 for 10 min and blocked in TBS with 0.5% BSA for 1 h. Primary antibodies, Krt14 (Abcam) and Krt10 (Covance), were diluted 1:1000 and incubated overnight at 4 °C. Secondary antibodies (Abcam, Life Technologies) were diluted 1:1000 and incubated at RT for 1 h. Images were acquired using a Keyence BZ-X700 fluorescent microscope. Fiji was used for image analysis.

**RNA fluorescent in situ hybridization (RNA-FISH)**. RNA-FISH was performed using the RNAscope Multiplex Fluorescent Detection Kit v1 according to manufacturer's instructions on fresh frozen 10 μM thick OCT sections. All sections were counterstained with ProLong Gold antifade reagent with DAPI. Images were acquired on a Leica SP8 confocal microscope. To ensure that images were comparable, they were all processed the same maximum intensity projection and brightness. Three biological replicates were analyzed for each FISH staining experiment.

**Ex vivo *Grhl3*$^{-/-}$ rescue experiment**. P0 WT and *Grhl3*$^{-/-}$ littermate dorsal back skin (12 mm in diameter, n = 3 each) was harvested as described above. Each piece of skin was cut in half and each half was incubated ex vivo in 100 nM Wnt-C59 small molecule inhibitor (R&D Systems, 5148) or DMSO, in 3 ml of EpiLife medium (ThermoFisher Scientific, MEPICF500) without calcium for 48 h at 37 °C. We analyzed twenty images from each biological replicate of each condition using Fiji to measure the thickness of the epidermis. We determined the average thickness of each biological replicate and used one tailed paired student's *t*-test to calculate the *p*-value.

**Sequencing**. Chromium Single Cell 3′ v1/v2 libraries were sequenced with either a Illumina HiSeq 2500 or a HiSeq4000 following the manufacturer's protocol.

**Primary computational analysis**. Raw sequencing data were demultiplexed and processed using Cellranger (10× Genomics version 2.01) using MM10 reference provided by 10× Genomics. The number of cells, mean reads per cell and median number of genes per cell for each experiments are listed in Supplementary Data 1. Cells were filtered and clustered using Seurat version 2.3.4[19]. For E14.5 and E16.5, cells with <600 and >4800 genes detected were removed. For P0, cells with <900 and >7700 genes detected were removed. For all time points, genes that are expressed in <6 cells were removed, and cells with >10% mitochondrial genes detected were removed. Raw gene-cell matrices were normalized and scaled. Percentage of mitochondrial genes, and the number of unique molecular identifiers (UMI) were regressed out using the RegressOut function. High variable genes were identified using a x.low.cutoff of 0.0125 and y.cutoff of 0.5 on mean variance dispersion plot. Data for each developmental age were integrated with canonical correlation analysis (CCA) on each biological replicate followed by alignment of subspace of each sample. Louvain clustering was then performed on the integrated samples. Single-cell consensus clustering (SC3) was carried out using $k = 2$ for basal IFE subclustering. Cell cycle analysis was carried out in Seurat using a list of cell cycle genes from the Regev laboratory[20].

We performed integrative analysis of developing IFE cells across E14.5, E16.5, and P0 using Seurat V3 (version 3.0.1) according to its tutorial (https://satijalab.org/seurat/v3.1/integration.html). Briefly, we performed SCTransform normalization separately for each time point, selected 3000 informative features using SelectIntegrationFeatures and PrepSCTIntegration functions, and performed integration analysis for joint clustering. Cells were visualized using UMAP (Uniform Manifold Approximation and Projection) algorithm[36]. For the differential gene expression analysis of IFE cells across E14.5, E16.5, and P0, we employed the following three criteria to identify enriched genes for each developmental stage[37]: (i) the *p*-values from Wilcoxon rank-sum tests are less than 0.01, (ii) the log fold-changes are higher than 0.25, and (iii) the percentage of expressed cells in the tested developmental stage is higher than 25%. Gene Ontology enrichment analysis of the enriched genes was performed and visualized using ClusterProfiler R package[38]. The code for the integration analysis is provided in the Supplementary Software file.

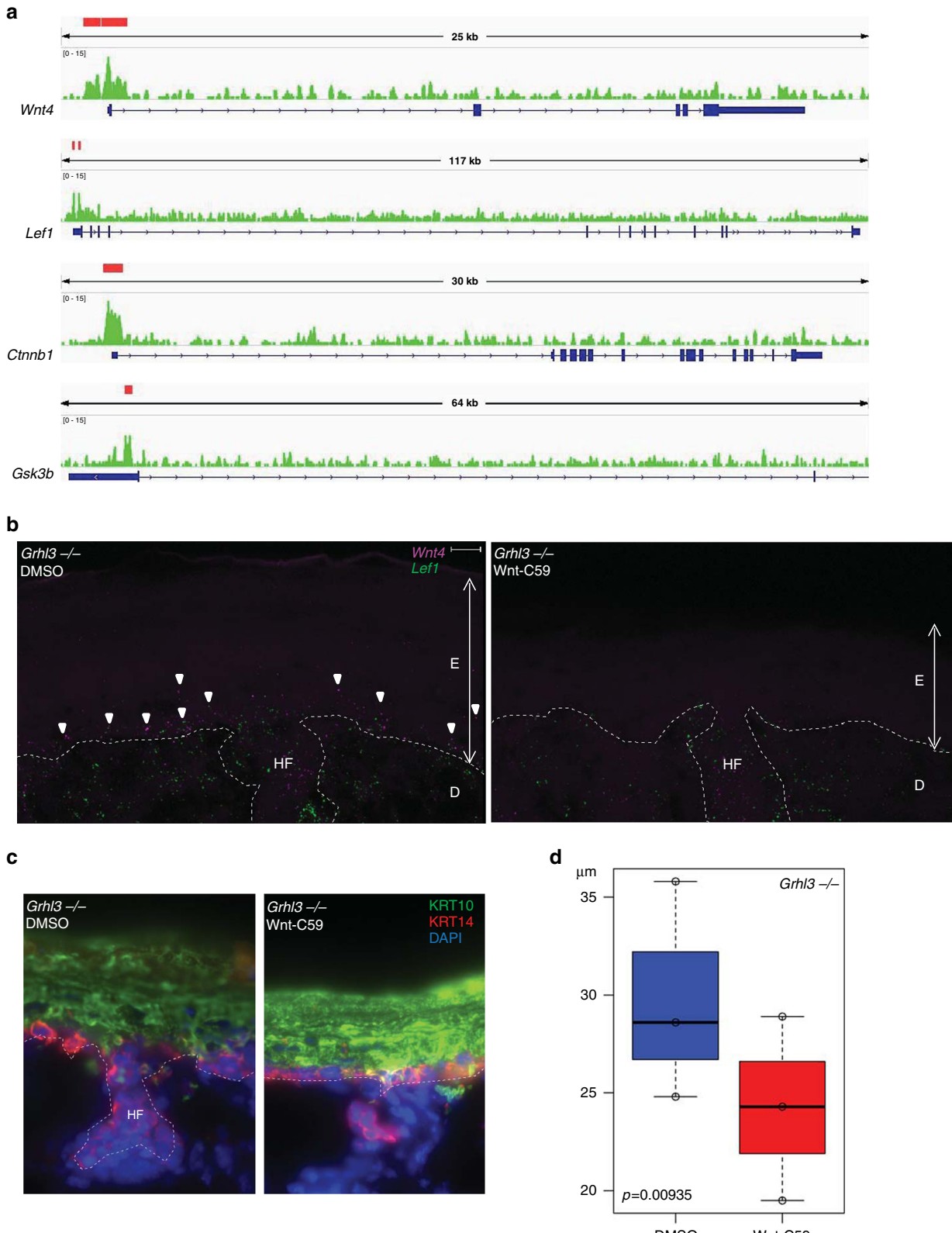

**Pseudotemporal trajectory and RNA-velocity analysis**. Monocle version 2.10.1 was used to construct differentiation trajectories of IFE using the highly variable genes identified from Seurat package as an ordering filter. DDRTree was used for dimension reduction. No root state is specified and ordering was done in an unsupervised manner. The *p*-value cutoff for identifying pseudotime-dependent genes was *p* < 0.01. To model gene expression changes in pseudotime, we used scEpath package[14], which first divides the pseudotime into ten equally spaced bins. Then the expression of each gene in each bin is estimated by the trimean of the expressions of this gene across all the cells located in this bin. Furthermore, scEpath smoothens the average expression of each gene using cubic regression splines. To study the temporal patterns and functional signatures of pseudotime-dependent genes, we performed k-medoids clustering based on the smoothed gene expression profiles, using Matlab function "kmedoids" with six clusters. We then created a heatmap to show the normalized-smoothed expression pattern of pseudotime-dependent genes. Genes within this heatmap were ordered such that nearest neighbors have similar expression profiles and genes within each cluster were

**Fig. 7 Wnt inhibition partially rescues epidermal hyperplasia in the Grhl3$^{-/-}$ IFE. a** GRHL3 ChIP-seq signal on the indicated Wnt signaling genes in the E16.5 epidermis. The red bars represent significant peaks. **b** Representative RNA-FISH images of *Wnt4* (pink) and *Lef1* (green) in *Grhl3$^{-/-}$* P0 epidermis treated ex vivo with control (DMSO; left panel) or Wnt inhibitor (Wnt-C59; right panel). There is decreased *Wnt4* and *Lef1* expression in the epidermis treated with the Wnt inhibitor. Arrowheads point to signal. $N = 3$ biologically independent samples each. White broken line traces basal lamina. E epidermis, D dermis, HF hair follicle. **c** Representative sections of *Grhl3$^{-/-}$* P0 epidermis treated ex vivo with control (DMSO; left panel) or Wnt inhibitor (Wnt-C59; right panel), representative image for $N = 3$ each. Krt14 staining, red; Krt10 staining, green; DAPI staining; blue; HF, hair follicle. There is decreased IFE thickening of the Wnt inhibitor treated epidermis. Scale bar $= 10\,\mu$m. **d** Summary statistics for IFE thickness, mean and standard error for ex vivo Wnt inhibitor experiments. *P*-value was calculated with one-tailed student *t*-test. $N = 3$ biologically independent samples each.

ordered according to expression peak. The average expression pattern of each cluster is calculated by the trimean of smoothed expressions of all the genes in that cluster. For the pseudotemporal trajectory analysis of developing IFE across E14.5, E16.5, and P0, we projected cells into a low-dimensional space using a diffusion-based manifold learning approach PHATE[39], with an input of the corrected data matrix from the integration analysis using Seurat V3. We then calculated the pseudotime of individual cells, identified the pseudotime-dependent genes and visualized pseudotemporal gene dynamics using scEpath package. Of note, the calculated pseudotime values were scaled ranging from 0 to 1. To discover key transcription factor programs responsible for cell states and state transitions during development, the transcription factors responsible for cell state transitions during epidermal differentiation were identified from the set of pseudotime-dependent genes based on the transcription factors annotated in AnimalTFDB[40].

RNA-velocity analysis was performed using the velocyto pipeline (https://github.com/velocyto-team/velocyto.R)[21]. The "run10x" function was used on Cellranger ouputs to generate the loom files using default parameters, mm10 gtf file provided by 10× Genomics and repeat mask gtf file from the UCSC Genome Browser. For the RNA velocity estimation, we used the standard R implementation of velocyto and only considered cells that were part of the pseudotime. The cell-to-cell distance matrix was calculated based on the PCA embedding. RNA velocity was estimated using gene-relative model with $k$-nearest neighbor cell pooling ($k = 20$). Embeddings from PCA and Monocle pseudotime analysis were used for velocity field projections.

**Reporting summary**. Further information on research design is available in the Nature Research Reporting Summary linked to this article.

## Data availability

The authors declare that all data supporting the findings of this study are available within the article and its Supplementary Information files or from the corresponding author upon reasonable request. The scRNA-seq data from this work have been deposited in the GEO database under the accession code GSE154579.

## Code availability

The code for Fig. 3 is available in the Supplementary Software file.

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

## Acknowledgements

We thank University of California, Irvine Genomic High-Throughput Facility for processing scRNA-seq libraries. This work was supported by NIH grants R01AR44882, U01AR073159, and P30AR075047 and the Irving Weinstein Foundation (to B.A.); and NSF grant DMS1763272, a grant from the Simons Foundation (594598), and NIH grant U01AR073159 (to Q.N.). Z.L. was supported by CA-T32 009054 from the National Cancer Institute.

## Author contributions

B.A. and Z.L. conceived the overall project. Z.L., J.C., Z.Li, Z.Q.L., and L.T. performed the experiments. ZL and J.C. curated the data. Z.L., S.J., Q.N., and B.A. analyzed the data. Z.L. wrote the manuscript with editorial input from all other authors.

## Competing interests

The authors declare no competing interests.
