## [Peer Review File · Nature Communications]

Reviewers' Comments:

Reviewer #1:

Remarks to the Author:

This manuscript addresses mechanisms of interfollicular epidermal (IFE) differentiation in mice by exploiting single-cell RNA sequencing (scRNA). The authors find that IFE differentiation is a single gradualistic process with a large number of transition cells between the basal and spinous layers. In addition, *Grhl3* contributes to the suppression of epidermal stem cell expansion and an abnormal stem cell state by repressing Wnt signaling.

Major points:

1) Variety in *Grhl3*-deficient IFE at P0.

The precise region of back skin is not mentioned in the Methods section. The dissected region for analysis should be more carefully described in terms of the antero-posterior axis level as well as the dorso-ventral level; for example, skin at the level of the 10th vertebra and so on. In addition, since *Grhl3*-deficient embryos display spina bifida, a description of the mutant phenotypes observed in the specimen is necessary.

More importantly, the authors analyzed two *Grhl3*-deficient embryos and found that they had different compositions of IFE to some extent (Fig.6B). Notably, IFE.A2 is observed only in the second of two embryos. Thus, the number of embryos subjected to scRNA-seq is not sufficient and at least P0 IFEs derived from five independent *Grhl3*-deficient embryos should be analyzed. Finally, they should examine whether all five RNA profiles support their model of Wnt signaling suppression by *Grhl3*.

2) Feasibility of their differentiation model (Fig.S8D).

The authors propose a revised model involving a "gradualistic process but not stepwise differentiation of IFE" in Fig.S8D. Their model is extracted from data analyzed only at P0 with RNA expression profiles. They should confirm their model by another experimental system such as the live cell imaging or cell lineage studies.

3) Feasibility of Wnt suppression by *Grhl3*.

The authors also propose that *Grhl3*-deficient IFE abnormalities are primarily due to the up-regulation of Wnt signaling. This statement is just speculation brought about by the only descriptive gene expression profiles. To demonstrate whether Wnt activation is actually involved in *Grhl3*-deficient phenotypes, several mouse genetic studies are necessary: for example, genetic interaction between mutant mice of *Grhl3* and Wnt-related genes in terms of IFE differentiation such as mutations of beta-catenin, TCF1 genes and so on and ii) overexpression of Wnt signaling can recapitulate IFE phenotypes similar to those of *Grhl3*-deficient embryos.

Moreover, further experimental demonstration how *Grhl3* can suppress Wnt signaling, i.e., the precise molecular mechanism underlying Wnt suppression, is essential.

Specific points:

i) Color codes indicating IFE.B.3 and IFFE.D are an almost identical shade of orange in Fig. 6, S6, 7 and S7. Most readers and reviewers are not able to distinguish between the two populations, even if the distribution patterns of IFE.B.3 are very crucial in this manuscript.

ii) Figures and text are inconsistent.

Fig.2D: Flg expression is described in the text but not shown in Fig. 2D.

Fig.4C: *Klf4*, *Tgm3* are described in the text but not shown in Fig. 4C.

Fig.6E: They mentioned "RNA-FISH experiments (Fig.6E)" but no Fig.6E is shown in this manuscript.

Reviewer #2:

Remarks to the Author:

In this manuscript Lin et al. utilizes 10x genomics- Single-Cell-RNA-Seq to investigate the role of GRHL3 in regulating epidermal stem cell transition states. The authors perform a large experiment by sequencing the epidermis of E14.5, E16.5, and P0 mouse skin from both wild type and *Ghrl3* null mice. *Ghrl3* null mice have an epidermal barrier phenotype, and the epidermal barrier is formed between E14.5 and E16.5, while fully maturing at P0. The authors utilize Seurat, Velocity, and Monocle to analyze the different time points and genetic variations in epidermal cell differentiation during development to conclude that *Ghrl3* is important in regulating IFE terminal differentiation. In essence this manuscript is a simple bioinformatic analysis of single cell RNA seq data with some validation of marker genes. The manuscript is underdeveloped at the experimental level and the computational analysis is premature. There are major concerns that prevent this reviewer from supporting publication of this manuscript. One issue is the premature nature of the computational analysis. The second issue is the complete lack of follow up studies to determine a molecular or cellular mechanism to understand how *Ghrl3* supports the development of an epidermal barrier.

Major:

- 1) The authors should be commended for utilizing up to date computational pipelines (i.e. velocity etc.). However, the organization of the data into 8 individual figures through re-analysis of sub-clusters, timing, and genetics is confusing/disjointed. To this reviewer, there seems to be two stories. The first story is the development of normal (WT) epidermis. To date, there does not appear to be an analysis at a single cell level how this occurs, and the authors dataset can provide novel insight into important process. The second story is the role of *Ghrl3* in establishment of the epidermal barrier. Consequently, I would highly recommends a collaborator in bioinformatics to tease out the critical points of both stories in order to take advantage of the nice datasets.
- 2) There is a lack of any mechanistic study at a cellular or molecular level to investigate the new findings gleaned from the Single Cell data. The manuscript is composed entirely of assaying single cell data. Could the authors generate initial experimental regimes exploring some aspects of the identified genes/pathways that they have identified in their datasets?

Reviewer #3:

Remarks to the Author:

The manuscript by Ziguang Lin et al. describes single-cell atlasing of epidermis in newborn mice and during embryonal development. The authors also study the role of a key transcription factor GRHL3 in epidermal differentiation using a knockout mouse model.

From both WT and KO mice, the authors report data from two biological replicates with mostly concordant results. The authors observe a continuous transcriptomic landscape underlying the canonical phenotypically distinct cell populations. The results suggest that GRHL3 regulates the basal proliferative stem compartment, and its role of GRHL3 in Wnt signaling is further established by ISH. Otherwise the paper presents limited functional or mechanistic validation of results.

However, the conclusions are reasonably well supported by the scRNA-seq data alone.

Altogether, the manuscript provides interesting new insights into the process of epidermal differentiation, which will hopefully be further expanded and also confirmed in human by future studies. The paper is also written very well and I quite enjoyed reading it.

I would recommend the paper for publication if the following concerns can be adequately addressed:

- Figure2: data from only one replicate is presented, while the two other P0 replicates are presented in Supplemental Fig 5. It would be informative to report how much cluster composition varies across these three individual biological replicates (as in Fig. 4D). The top marker genes (and their p-values) for each cluster should also be included as supplementary data (or at least

highlighted in 2B) to provide the rationale for cluster annotation.

- Page 3: First results are already presented in the introduction section: "To better understand epidermal differentiation, we investigated 85,286 single cell transcriptomes from mouse skin at embryonic day (E) 14.5, E16.5, and P0 in wild type (WT) and *Grhl3*^{-/-} mice (Figure 1A). " – I found this somewhat confusing and in my opinion, it would be more logical to move this to the beginning of the results section.
- Page 4: "38,879 mean number of reads per cell and 2,388 mean number of genes per cell " - these details should be presented for all scRNA-seq datasets (e.g. in the methods section)
- Figure 4C: No statistical test is presented to support the statement that *Klf4*, *Tgm3* and *Lor* are markedly reduced in the KO
- Figure 5 Legend "Figure 5. scRNA-seq reveals decreased size of proliferating transition cells and progress to terminal differentiation in the E16.5 IFE. " In my opinion, this is somewhat inaccurate as it can be understood as referring to physical size of the cells instead of the size of the population, which I presume is what the authors mean.
- Figures 6A & 7A : nearly identical colours are used for IFE.A.1 and IFE.D making the plots very difficult to interpret
- Figures 2C and 7A(left) seem somewhat redundant. Is there any conceptual difference between the pseudotime analyses presented in them?
- Figure 6A and corresponding text: IFE.A2 cells are found almost exclusively in *Grhl3*^{-/-}2. Why is this the case?
- Color key is missing from many plots
- Figures 2E, 3D, 6D, 8B: no scale bar is shown
- A statement on availability of data should be included

Response to Referees:

We thank all three reviewers for their careful reading of our work, and for providing valuable suggestions and insights. Below we outline our specific responses to their comments and concerns. In the revised manuscript, we highlight major changes in green.

Responses to Reviewer #1

This manuscript addresses mechanisms of interfollicular epidermal (IFE) differentiation in mice by exploiting single-cell RNA sequencing (scRNA). The authors find that IFE differentiation is a single gradualistic process with a large number of transition cells between the basal and spinous layers. In addition, *Grhl3* contributes to the suppression of epidermal stem cell expansion and an abnormal stem cell state by repressing Wnt signaling.

Major points:

1) Variety in *Grhl3*-deficient IFE at P0.

The precise region of back skin is not mentioned in the Methods section. The dissected region for analysis should be more carefully described in terms of the antero-posterior axis level as well as the dorso-ventral level; for example, skin at the level of the 10th vertebra and so on. In addition, since *Grhl3*-deficient embryos display spina bifida, a description of the mutant phenotypes observed in the specimen is necessary.

More importantly, the authors analyzed two *Grhl3*-deficient embryos and found that they had different compositions of IFE to some extent (Fig.6B). Notably, IFE.A2 is observed only in the second of two embryos. Thus, the number of embryos subjected to scRNA-seq is not sufficient and at least P0 IFEs derived from five independent *Grhl3*-deficient embryos should be analyzed. Finally, they should examine whether all five RNA profiles support their model of Wnt signaling suppression by *Grhl3*.

Response: We now include the requested description in the Methods. All *Grhl3*^{-/-} embryos displayed spina bifida. We removed approximately 6 mm in diameter circular sample of backskin, centered at the midline in one axis and on a line drawn between the forelimbs in the other axis. This location avoids skin near the spina bifida defect. We used the same dissected region previously to study skin gene expression in *Grhl3*^{-/-} embryos (Yu et al, Dev Biol, 2006).

As to the IFE.A2 population only being present in one of two *Grhl3*^{-/-} P0 embryos, we apologize for leaving this impression. This population is present in both P0 embryos, albeit in greater numbers in one of the embryos (135 vs. 18) (Fig. 4B); we now make this point clearly in the text (top of page 8). We do not believe that this variability calls for greater number of replicates, especially as the IFE.A2 population is not a focus of our study. Furthermore, our study is already based on the analysis of 7 WT (2 E14.5; 2 E16.5; and 3 E18.5/P0) and 6 *Grhl3*^{-/-} embryos (2 E14.5; 2 E16.5; and 2 E18.5/P0), using littermate controls. The markers for IFE.A2 cells (Supplemental Table 1; tab for Supplemental Fig. 5D, 6D) show that these cells express relatively high levels of mitochondrial genes, an indicator of cellular stress and possible progress toward cell death. Even on a pure C57/Bl6 background there is some variability in the severity of

the epidermal phenotype of the *Grhl3*^{-/-} mice and we speculate that the IFE.A2 population may be a sensitive marker of the cellular damage associated with hyperproliferation and defective barrier.

As to the Wnt signaling suppression by GRHL3, we have validated this with additional markers in the scRNA-seq data (Fig. 6A, B, Supplemental Fig. 8A). But more directly in response to the reviewer's suggestion, this observation is now validated in additional embryos, using RNA-FISH for *Wnt4*, *Lef1*, and *Dkk1*, showing that the results are consistent in multiple biological replicates and observed with a method independent of scRNA-seq (Fig. 6C, D). Furthermore, the revised manuscript now features GRHL3 ChIP-seq experiments showing direct binding to promoters of Wnt signaling genes (Fig. 7A) and functional rescue experiments, supporting the biological importance Wnt signaling suppression by GRHL3 (Fig. 7B, C, D).

2) Feasibility of their differentiation model (Fig.S8D).

The authors propose a revised model involving a “gradualistic process but not stepwise differentiation of IFE” in Fig.S8D. Their model is extracted from data analyzed only at P0 with RNA expression profiles. They should confirm their model by another experimental system such as the live cell imaging or cell lineage studies.

Response: The reviewer homes in on gradualistic differentiation, one of the most interesting observations of our study. The IFE differentiation lineage is linear without branches, and therefore it is not so clear how one could use cell lineage studies to demonstrate this phenomenon. On the other hand, the scRNA-seq technique, which detects the whole transcriptome rather than a single marker or a collection of few markers, provides an unbiased approach to monitor the IFE differentiation process at a global scale.

Please note that the gradualistic feature of IFE differentiation was consistently observed when several different unsupervised computational approaches were applied to the datasets. First, the gradualistic feature was seen when projecting cells into a low-dimensional space using four different dimension-reduction algorithms, including tSNE (Fig. 1A), DDRTree (Fig. 1C), UMAP (Fig. 3A) and PHATE (Fig. 3D). Second, the gradualistic feature was seen when analyzing the pseudotemporal dynamics during IFE differentiation. At P0, a gradualistic change of the transcriptomic dynamics was seen during IFE differentiation by investigating the transcriptomic profiles of 4,292 pseudotime-dependent genes (Fig. 1F). We also observed this when performing integrative analysis of scRNA-seq data across three developmental stages, E14.5, E16.5, and E18.5/P0 (Fig. 3A, D, F). The integrative analysis indicates the gradualistic differentiation is already evident as early as E14.5. Third, RNA velocity analysis, which predicts cell fate solely based on pre-mRNA and mature mRNA species of the same transcript, consistently revealed the gradualistic change of RNA dynamics across different replicates (Fig. 5C).

Furthermore, we demonstrate transition cells during IFE differentiation, using experimental systems distinct from scRNA-seq, namely mRNA and protein localization studies (Fig. 1E, 4D, Supplemental Fig. 6C). We agree with the reviewer that aspects of gradualistic differentiation could be observed with other experimental approaches such as live cell imaging. While live cell imaging of transgenic mice with fluorescent K14 and K10 reporters would detect transition cells and give greater time information, it would not provide fundamentally different information from our static detection of transition cells. It is not, however, feasible to establish this system and conduct such experiments during the revision period, especially given the challenging Covid-19 environment.

We hope that the reviewer appreciates the strength of the data, combining several independent unsupervised computational approaches with mRNA and protein localization studies of key markers. We have added the reviewers point about other approaches to the Discussion in the revised manuscript.

3) Feasibility of Wnt suppression by Grhl3.

The authors also propose that Grhl3-deficient IFE abnormalities are primarily due to the up-regulation of Wnt signaling. This statement is just speculation brought about by the only descriptive gene expression profiles. To demonstrate whether Wnt activation is actually involved in Grhl3-deficient phenotypes, several mouse genetic studies are necessary: for example, genetic interaction between mutant mice of Grhl3 and Wnt-related genes in terms of IFE differentiation such as mutations of beta-catenin, TCF1 genes and so on and ii) overexpression of Wnt signaling can recapitulate IFE phenotypes similar to those of Grhl3-deficient embryos. Moreover, further experimental demonstration how Grhl3 can suppress Wnt signaling, i.e., the precise molecular mechanism underlying Wnt suppression, is essential.

Response: We thank the reviewer for these suggestions. Previously, it was thought that the Wnt pathway was important for hair follicle stem cells but not for the function of IFE stem cells. This idea changed with work from the Nusse (Lim et al, Science, 2013) and Millar (Choi et al, Cell Stem Cell, 2013) laboratories, showing that autocrine Wnt activity does indeed drive IFE stem cell proliferation and renewal. Prior work from the Fuchs laboratory (Gat et al, Cell, 1998) showed that constitutive active beta catenin in K14-expressing epidermal cells led to increased IFE thickness in back skin, in addition to hair follicle neogenesis. Thus, there is strong evidence in the literature that gain and loss of Wnt activity, respectively, stimulates and inhibits expansion of IFE stem cells.

The reviewer is correct: we based the initial idea about GRHL3 suppressing Wnt activity on descriptive gene expression profiles from our scRNA-seq data, in addition to inference from the studies mentioned above. But in response to the reviewer's suggestions, we have obtained additional experimental support. *I*) RNA-FISH of Wnt4 and Lef1, a downstream effector of Wnt pathway, show that both transcripts are significantly increased in the *Grhl3*^{-/-} IFE (Fig. 6C). These experiments show that indicators of increased Wnt signaling in *Grhl3*^{-/-} epidermis are

observed by methods distinct from scRNA-seq. 2) We have shown the functionality of the increased Wnt signaling in ex vivo experiments where a small molecule inhibitor of Wnt ligand secretion significantly countered the increased epidermal thickness of the *Grhl3*^{-/-} IFE (Fig. 7B, C, D). These experiments suggest that the increased epidermal thickness in the *Grhl3*^{-/-} epidermis is at least partially driven by Wnt activity. 3) We analyzed our earlier GRHL3 ChIP-seq data from E16.5 mouse skin (Gordon et al, J Clin Invest, 2015) and found that GRHL3 directly binds to promoter regions of several Wnt signaling genes, including *Wnt4*, *Lef1*, *Ctnnb1*, and *Gsk3b* (Fig. 7A). We have previously shown (Klein et al, PLOS Genet, 2015) that promoter binding of GRHL3 is often associated with repression. These data suggest that GRHL3 may directly represses Wnt pathway genes. 4) With both scRNA-seq and RNA-FISH, we show that *Dkk11*, a known Wnt ligand antagonist is expressed to high levels in the spinous layer but not in the basal layer. *Dkk11* expression level, as well as number of cells expressing *Dkk11*, are significantly reduced in the *Grhl3*^{-/-} IFE (Fig. 6B, D, Supplemental Fig. 8C), suggesting that loss of Wnt inhibitory signaling from the differentiated IFE compartment can also contribute to the Wnt overactivity in the *Grhl3*^{-/-} epidermis.

Specific points:

i) Color codes indicating IFE.B.3 and IFFE.D are an almost identical shade of orange in Fig. 6, S6, 7 and S7. Most readers and reviewers are not able to distinguish between the two populations, even if the distribution patterns of IFE.B.3 are very crucial in this manuscript.

Response: We have improved our selection of colors to make them more discriminatory.

ii) Figures and text are inconsistent.

Fig.2D: Flg expression is described in the text but not shown in Fig. 2D.

Response: Previous Fig. 2D is currently Fig. 1D. Flg has the same pattern as Lor in our data but since it is not shown in the figure, we have removed the reference to this marker in the text.

Fig.4C: *Klf4*, *Tgm3* are described in the text but not shown in Fig. 4C.

Response: With the integrated analysis of developmental time points, this figure has been removed and these markers are not currently mentioned in this context.

Fig.6E: They mentioned “RNA-FISH experiments (Fig.6E)” but no Fig.6E is shown in this manuscript.

Response: We apologize for this mistake; we have corrected it.

Responses to Reviewer #2

In this manuscript Lin et al. utilizes 10x genomics- Single-Cell-RNA-Seq to investigate the role of GRHL3 in regulating epidermal stem cell transition states. The authors perform a large

experiment by sequencing the epidermis of E14.5, E16.5, and P0 mouse skin from both wild type and *Ghrl3* null mice. *Ghrl3* null mice have an epidermal barrier phenotype, and the epidermal barrier is formed between E14.5 and E16.5, while fully maturing at P0. The authors utilize Seurat, Velocity, and Monocle to analyze the different time points and genetic variations in epidermal cell differentiation during development to conclude that *Ghrl3* is important in regulating IFE terminal differentiation. In essence this manuscript is a simple bioinformatic analysis of single cell RNA seq data with some validation of marker genes. The manuscript is underdeveloped at the experimental level and the computational analysis is premature. There are major concerns that prevent this reviewer from supporting publication of this manuscript. One issue is the premature nature of the computational analysis. The second issue is the complete lack of follow up studies to determine a molecular or cellular mechanism to understand how *Ghrl3* supports the development of an epidermal barrier.

Major:

1) The authors should be commended for utilizing up to date computational pipelines (i.e. velocity etc.). However, the organization of the data into 8 individual figures through re-analysis of sub-clusters, timing, and genetics is confusing/disjointed. To this reviewer, there seems to be two stories. The first story is the development of normal (WT) epidermis. To date, there does not appear to be an analysis at a single cell level how this occurs, and the authors dataset can provide novel insight into important process. The second story is the role of *Ghrl3* in establishment of the epidermal barrier. Consequently, I would highly recommends a collaborator in bioinformatics to tease out the critical points of both stories in order to take advantage of the nice datasets.

Response: We thank the reviewer for these comments and suggestions. In this revision, we worked closely with Suoqin Jin, the second author, and Qing Nie, the co-corresponding author, who are highly experienced and well published in the analysis of scRNA-seq experiments. We have gone to extensive lengths to address these concerns by performing the suggested analysis of E14.5, E16.5 and P0 samples in an integrative fashion, and by comparing the epidermal development and epidermal differentiation analyses (new Fig. 3; previous Figs. 4 and 5 were removed). Additions to the revised manuscript include: *1*) Joint clustering of IFE cells from WT E14.5, E16.5 and P0 samples to understand how the cellular composition changes during epidermal development from E14.5 to P0. We found that the proportion of IFE basal cells (in particular the proliferative basal cells) decreased, and the proportion of IFE transition cells and differentiated cells increased from E14.5 to P0 (Fig. 3A, B, C). *2*) Joint pseudotime analysis of IFE cells from WT E14.5, E16.5 and P0 IFEs to understand IFE differentiation in the context of epidermal development from E14.5 to P0. This has made the overall presentation of the data more cohesive and clearly shows that the IFE differentiation program is already operating as early as E14.5 (Fig. 3D, E, F). *3*) GO enrichment analysis of differentially expressed genes at E14.5, E16.5, and P0 revealed enriched transcriptomic signatures at each developmental stage, including in specific populations (Fig. 3G, H, I). *4*) We have also made clearer in the text, the distinction between development on one hand and the formation of the epidermal barrier and the role of *GRHL3* in differentiation on the other. In part, we have done this by rearranging the order of presentation, starting with our analysis of the P0 IFE, which introduces the main cell states

and concepts, then showing how the IFE develops during embryogenesis, ending with testing how GRHL3 regulates IFE differentiation and cellular composition of the IFE. We have also added a clearer transition sentence when we move to the developmental analysis (page 6).

2) There is a lack of any mechanistic study at a cellular or molecular level to investigate the new findings gleaned from the Single Cell data. The manuscript is composed entirely of assaying single cell data. Could the authors generate initial experimental regimes exploring some aspects of the identified genes/pathways that they have identified in their datasets?

Response: We have performed *ex vivo* experiments showing that blocking Wnt secretion rescues the IFE hyperplasia in the *Grhl3*^{-/-} IFE (Fig. 7). This experiment suggests that GRHL3-suppression of Wnt signaling is functionally important. Analysis of our previous GRHL3 ChIP-seq data indicates that Wnt pathway genes, including *Wnt4* and *Lef1*, are direct targets of GRHL3 (Fig. 7A). Furthermore, we characterize the epidermal expression of Wnt inhibitor *Dkk1* and show its decreased expression in the *Grhl3*^{-/-} IFE (Fig. 6B, D, Supplemental Fig. 8C). We have also performed additional RNA-FISH experiments to show up-regulation of Wnt signaling genes in the *Grhl3*^{-/-} IFE (Fig. 6C). See also response to point 3 of Reviewer #1.

Responses to Reviewer #3

The manuscript by Ziguang Lin et al. describes single-cell atlasing of epidermis in newborn mice and during embryonal development. The authors also study the role of a key transcription factor GRHL3 in epidermal differentiation using a knockout mouse model.

From both WT and KO mice, the authors report data from two biological replicates with mostly concordant results. The authors observe a continuous transcriptomic landscape underlying the canonical phenotypically distinct cell populations. The results suggest that GRHL3 regulates the basal proliferative stem compartment, and its role of GRHL3 in Wnt signaling is further established by ISH. Otherwise the paper presents limited functional or mechanistic validation of results. However, the conclusions are reasonably well supported by the scRNA-seq data alone. Altogether, the manuscript provides interesting new insights into the process of epidermal differentiation, which will hopefully be further expanded and also confirmed in human by future studies. The paper is also written very well and I quite enjoyed reading it.

We thank the reviewer for the positive assessment.

I would recommend the paper for publication if the following concerns can be adequately addressed:

- Figure 2: data from only one replicate is presented, while the two other P0 replicates are presented in Supplemental Fig 5. It would be informative to report how much cluster composition varies across these three individual biological replicates (as in Fig. 4D). The top marker genes (and their p-values) for each cluster should also be included as supplementary data (or at least highlighted in 2B) to provide the rationale for cluster annotation.

Response: We performed the initial experiment, which we analyzed extensively, with a single WT P0 epidermis. We feature the data from this experiment in Figs. 1 and 2 and in Supplemental Fig. 1 and 2. The initial definition of cell states and the idea of gradualistic differentiation came from these analyses. As part of the development study and the GRHL3 experiments, we analyzed two additional WT P0 epidermis, which we mainly feature in Figs. 3-5 and in Supplemental Figs. 5-7. The clustering results are consistent between the two sets of experiments and between the two WT replicates that were analyzed in parallel in the later set. The only difference is that with the later WT P0 samples, we had greater number of cells and two additional clusters emerged: IFE.B3, which are cycling basal cells (these were distributed among the IFE.B1 and IFE.B2 clusters in Fig. 1), and IFE.TD, which are terminally differentiated cells (these were part of the IFE.D cluster in Fig. 1). We describe these differences in the text.

We now provide all gene lists as an Excel document.

- Page3: First results are already presented in the introduction section: “To better understand epidermal differentiation, we investigated 85,286 single cell transcriptomes from mouse skin at embryonic day (E) 14.5, E16.5, and P0 in wild type (WT) and *Grhl3*^{-/-} mice (Figure 1A).” – I found this somewhat confusing and in my opinion, it would be more logical to move this to the beginning of the results section.

Response: We agree with the reviewer and have moved the description of specific findings to the Results.

- Page 4: “38,879 mean number of reads per cell and 2,388 mean number of genes per cell” - these details should be presented for all scRNA-seq datasets (e.g. in the methods section)

Response: These details are now included for all datasets as suggested by the reviewer.

- Figure 4C: No statistical test is presented to support the statement that *Klf4*, *Tgm3* and *Lor* are markedly reduced in the KO

Response: We removed previous Fig. 4 and replaced with the integrated analysis of all three developmental time points (current Fig. 3); the statement has therefore been removed from the paper.

- Figure 5 Legend “Figure 5. scRNA-seq reveals decreased size of proliferating transition cells and progress to terminal differentiation in the E16.5 IFE.” In my opinion, this is somewhat inaccurate as it can be understood as referring to physical size of the cells instead of the size of the population, which I presume is what the authors mean.

Response: We agree, but when we removed previous Fig. 5 and replaced with the integrated analysis of all three time-points (current Fig. 3), we also removed the sentence with this ambiguity.

- Figures 6A & 7A : nearly identical colours are used for IFE.A.1 and IFE.D making the plots very difficult to interpret

Response: We have made all colors more discriminatory.

- Figures 2C and 7A (left) seem somewhat redundant. Is there any conceptual difference between the pseudotime analyses presented in them?

Response: Currently, these are Fig. 1C and Fig. 5A. The reviewer is correct that there is no conceptual difference between the pseudotime analyses presented in them. We prefer, however, to keep both for the following reasons: 1) Fig. 1C represents a single replicate, Fig. 5A represents two replicates; together all three are a powerful demonstration of the consistency in the data. 2) The pseudotime in Fig. 1C corresponds to the data used in the rest of the analysis in Fig. 1 and Fig. 2 and is therefore the most appropriate comparative data for these analyses. 3) The data in Fig. 5A represents the littermate controls of the *Grhl3*^{-/-} IFE and therefore is a more appropriate control for this experiment.

- Figure 6A and corresponding text: IFE.A2 cells are found almost exclusively in *Grhl3*^{-/-}2. Why is this the case?

Response: We agree with the reviewer that although we find the IFE.A2 cells in both replicates of *Grhl3*^{-/-} IFE, this population is more abundant in one of the replicates (135 vs. 18). We do not know why this population shows variability although we know that the severity of the epidermal phenotype of *Grhl3*^{-/-} mice varies somewhat although the mice are on a pure C57Bl6 background. We determined the top marker gene expression for the IFE.A2 population, finding that these cells maybe undergoing stress. These cells may reflect a more severe barrier abnormality. We discuss these results in the revised manuscript (pages 8 and 9). See also response to point 1 of Reviewer #1.

- Color key is missing from many plots

Response: We added color keys to all plots.

- Figures 2E, 3D, 6D, 8B: no scale bar is shown

Response: We added scale bars to all images.

- A statement on availability of data should be included

Response: This statement is included; we are in the process of depositing the data to GEO. We also plan to make the data accessible in a user-friendly format in the public data portal of the UCI Skin Center web site: <http://www.skingenes.net/>

References:

- Yu, Z., Lin, K. K., Bhandari, A., Spencer, J. A., Xu, X., Wang, N., Lu, Z., Gill, G. N., Roop, D. R., Wertz, P., and Andersen, B. (2006) The Grainyhead-like epithelial transactivator Get-1/Grhl3 regulates epidermal terminal differentiation and interacts functionally with LMO4. *Dev Biol* **299**, 122-136
- Lim, X., Tan, S. H., Koh, W. L., Chau, R. M., Yan, K. S., Kuo, C. J., van Amerongen, R., Klein, A. M., and Nusse, R. (2013) Interfollicular epidermal stem cells self-renew via autocrine Wnt signaling. *Science* **342**, 1226-1230
- Choi, Y. S., Zhang, Y., Xu, M., Yang, Y., Ito, M., Peng, T., Cui, Z., Nagy, A., Hadjantonakis, A. K., Lang, R. A., Cotsarelis, G., Andl, T., Morrisey, E. E., and Millar, S. E. (2013) Distinct functions for Wnt/beta-catenin in hair follicle stem cell proliferation and survival and interfollicular epidermal homeostasis. *Cell Stem Cell* **13**, 720-733
- Gat, U., DasGupta, R., Degenstein, L., and Fuchs, E. (1998) De Novo hair follicle morphogenesis and hair tumors in mice expressing a truncated beta-catenin in skin. *Cell* **95**, 605-614.
- Gordon, W. M., Zeller, M. D., Klein, R. H., Swindell, W. R., Ho, H., Espetia, F., Gudjonsson, J. E., Baldi, P. F., and Andersen, B. (2014) A GRHL3-regulated repair pathway suppresses immune-mediated epidermal hyperplasia. *J Clin Invest* **124**, 5205-5218
- Klein, R. H., Lin, Z., Hopkin, A. S., Gordon, W., Tsoi, L. C., Liang, Y., Gudjonsson, J. E., and Andersen, B. (2017) GRHL3 binding and enhancers rearrange as epidermal keratinocytes transition between functional states. *PLoS Genet* **13**, e1006745

Reviewers' Comments:

Reviewer #1:

Remarks to the Author:

The authors have newly conducted rescue experiments with the chemical inhibitor of Wnt and include new information regarding GRHL3 Chip-seq data for Wnt suppression. These additional findings clarified many of the major issues I concerned. I feel the revised manuscript was improved.

Reviewer #2:

Remarks to the Author:

In my initial review of this manuscript of by Lin et al. I had two main concerns. The first was the premature nature of the computational analysis of the single cell RNA-seq data. The second was the lack of follow up in vivo studies to verify the a cellular and molecular mechanism. In response the authors have generated a large amount of new computational data by collaborating with Dr. Nie. These analysis greatly improve the manuscript. My second concern was addressed by the authors analyzing ChIP-seq experiments for GRHL3 which suggests that it associates with Wnt signaling elements such as Axin, Gsxk3b and others. These additions improve the manuscript, support the hypothesis, and address my concerns.

Reviewer #3:

Remarks to the Author:

The manuscript by Ziguang Lin et al. presents a single-cell analysis of interfollicular epidermal differentiation. Authors show that despite defined cellular boundaries at the spatial level, the differentiation is a fundamentally gradual process at molecular level. The data also reveals distinct basal stem cell states and identifies GRHL3 as a novel regulator. In the revised manuscript, the authors have also further expanded on the role of GRHL3 as a regulator of Wnt signalling. These are important findings and the analyses are technically sound. In the revised manuscript, the authors have addressed the concerns I raised upon the initial submission, and I have no further reservations about this manuscript.

Reviewer #1 (Remarks to the Author):

The authors have newly conducted rescue experiments with the chemical inhibitor of Wnt and include new information regarding GRHL3 Chip-seq data for Wnt suppression. These additional findings clarified many of the major issues I concerned. I feel the revised manuscript was improved.

--

Reviewer #2 (Remarks to the Author):

In my initial review of this manuscript of by Lin et al. I had two main concerns. The first was the premature nature of the computational analysis of the single cell RNA-seq data. The second was the lack of follow up in vivo studies to verify the a cellular and molecular mechanism. In response the authors have generated a large amount of new computational data by collaborating with Dr. Nie. These analysis greatly improve the manuscript. My second concern was addressed by the authors analyzing ChIP-seq experiments for GRHL3 which suggests that it associates with Wnt signaling elements such as Axin, Gsxk3b and others. These additions improve the manuscript, support the hypothesis, and address my concerns.

--

Reviewer #3 (Remarks to the Author):

The manuscript by Ziguang Lin et al. presents a single-cell analysis of interfollicular epidermal differentiation. Authors show that despite defined cellular boundaries at the spatial level, the differentiation is a fundamentally gradual process at molecular level. The data also reveals

distinct basal stem cell states and identifies GRHL3 as a novel regulator. In the revised manuscript, the authors have also further expanded on the role of GRHL3 as a regulator of Wnt signalling.

These are important findings and the analyses are technically sound. In the revised manuscript, the authors have addressed the concerns I raised upon the initial submission, and I have no further reservations about this manuscript.

Again, we thank you and the reviewers for improving the manuscript.